# Parabrachial tachykinin1-expressing neurons involved in state-dependent breathing control

Joseph W. Arthurs[1,2], Anna J. Bowen [1], Richard D. Palmiter [1] &
Nathan A. Baertsch [2,3] ✉

Breathing is regulated automatically by neural circuits in the medulla to maintain homeostasis, but breathing is also modified by behavior and emotion. Mice have rapid breathing patterns that are unique to the awake state and distinct from those driven by automatic reflexes. Activation of medullary neurons that control automatic breathing does not reproduce these rapid breathing patterns. By manipulating transcriptionally defined neurons in the parabrachial nucleus, we identify a subset of neurons that express the *Tac1*, but not *Calca*, gene that exerts potent and precise conditional control of breathing in the awake, but not anesthetized, state via projections to the ventral intermediate reticular zone of the medulla. Activating these neurons drives breathing to frequencies that match the physiological maximum through mechanisms that differ from those that underlie the automatic control of breathing. We postulate that this circuit is important for the integration of breathing with state-dependent behaviors and emotions.

Most sensorimotor functions of the nervous system are strongly gated by neural state. This includes rhythmic motor behaviors driven by central pattern generators (CPGs) such as walking, swimming, flying, and chewing[1,2]. These rhythmic motor outputs are conditionally active in the awake state but are largely absent during sleep or anesthesia. Breathing, however, is a vital rhythmic motor output that must continuously function to ensure homeostasis despite changes in state. This so called "automatic" respiratory control must be robust yet sufficiently flexible to reflexively tune breathing patterns in response to changes in metabolic or environmental demands.

Yet, in addition to its automatic regulation, breathing is conditionally modified by behavior and emotion. This state-dependent control yields breathing patterns in awake animals that are much more complex than the regular repeated process of inspiration and expiration that remains during sleep or anesthesia. For example, in humans breathing is dramatically and differentially altered by laughter, crying, fear, and anger as well as volitional control, vocalizations, and other uniquely human behaviors such as playing wind instruments[3–7]. In many mammals including humans, breathing patterns in the awake state are conditionally modified for sensory exploration in the form of sniffing[8,9]. In rodents, rapid breathing patterns related to sniffing are common in adults but are mostly absent in neonates and follow a developmental trajectory that is distinct from normal breathing[10]. In adults, the transition from resting breathing to sniffing can be abrupt, sometimes occurring within a single respiratory cycle, and can cease just as rapidly[11,12]. During sniffing episodes, breathing can be accelerated to frequencies as high as ~12 Hz that are precisely tuned to regulate olfactory processing[13,14]. Similar rapid breathing patterns are observed in response to startling, fearful, or painful stimuli[15–18]. Thus, breathing patterns in awake freely behaving adult rodents are highly dynamic and are characterized by low- and high-frequency modes with distinct functional roles.

The preBötzinger Complex (preBötC), a small region located bilaterally in the ventrolateral medulla, is sufficient to produce a rhythm that drives respiratory motor activity[19] and is widely accepted as the rhythmogenic core of the respiratory CPG[20–24]. The inexorable

[1]Howard Hughes Medical Institute and Department of Biochemistry, University of Washington, Seattle, WA 98195, USA. [2]Center for Integrative Brain Research, Seattle Children's Research Institute, Seattle, WA 98101, USA. [3]Pulmonary Critical Care and Sleep Medicine, Department of Pediatrics, University of Washington, Seattle, WA, USA. ✉e-mail: nathan.baertsch@seattlechildrens.org

and automatic properties of this rhythm-generating network have provided a unique opportunity to study the generation of rhythmic activity in anesthetized or in vitro vertebrate preparations. Accordingly, extensive research has leveraged these experimental preparations to unravel mechanisms of respiratory control that underlie its critical physiological role in maintaining homeostasis[25–27]. Although the necessity of the preBötC for normal, automatic breathing is clear[28–31], its role in conditional and state-dependent respiratory control is less well established. Retrograde tracing experiments have revealed that the preBötC receives numerous inputs from regions throughout the brain including cortex, hypothalamus, central amygdala, substantia nigra, pariaqueductal gray, red nucleus, and dorsolateral pons[32–34]; however, the functional roles for most of these preBötC inputs have not been investigated. Due to the apparent necessity of the preBötC for all inspiratory behaviors[28], but also see[35], it is generally assumed that changes in these inputs modulate a single rhythm-generating mechanism in the preBötC to adapt breathing to behavior and emotion as well as changes in physiological demands. However, optogenetic activation of preBötC neurons in awake or anesthetized mice has not reproduced the rapid breathing patterns that characterize the awake state, only driving breathing frequencies up to ~6 Hz[36–38], an increase comparable to that elicited by near-maximal chemoreflex stimulation[39,40]. Thus, the circuits and mechanisms that drive the rapid breathing patterns that exemplify the state-dependent control of breathing in awake mice remain unresolved.

An important role of the parabrachial nucleus (PBN) in the dorsolateral pons for respiratory regulation has been appreciated for decades[41,42]. However, defining its specific contributions to rhythm generation and modulation has been difficult. This is due, in part, to the large variety of cell types in this region that are diverse in their gene expression, axonal projections, and functions[43–48], which has hindered interpretation of non-cell-type specific manipulation experiments. Indeed, the PBN is considered an integrative hub for many behaviors, affective states, and autonomic responses[49,50] and contains ~12 transcriptionally distinct neuronal populations as defined by RNA-Sequencing[48]. However, modern neuroscience approaches that allow manipulations of transcriptionally and anatomically defined cell-types have only recently begun to be applied to unravel how restricted PBN populations may regulate breathing[17,51].

Here, we implement an intersectional genetic strategy to identify a specific subpopulation of neurons in the lateral PBN characterized by the expression of tacykinin1 (Tac1, encoded by the *Tac1* gene), but not calcitonin gene-related peptide (CGRP, encoded by the *Calca* gene), that drives state-dependent breathing patterns via projections to the preBötC and surrounding ventral intermediate reticular zone of the medulla (vIRt). In the awake state, these neurons potently drive rapid breathing patterns that phenocopy those produced spontaneously by awake freely behaving mice. However, under light anesthesia, activation of this circuit no longer affects breathing, suggesting it plays a minimal role in its automatic regulation. This state-dependent control of breathing exhibits characteristics that are distinct from the well-characterized properties of rhythm generation that underlie its automatic control. These results significantly expand our basic understanding of how breathing is automatically regulated to ensure homeostasis, while remaining highly modifiable in the awake state to allow for its conditional integration with behavior and emotion.

## Results
### Rapid and dynamic breathing patterns are a specific feature of the awake normally behaving state in mice
To better characterize breathing patterns driven by mechanisms of automatic and state-dependent control, we used whole-body plethysmography to record respiratory activity in unrestrained adult C57BL/6J mice[52] (Fig. 1a). Breathing patterns in the awake freely behaving state were measured under control (room air) conditions and

were compared to breathing patterns during a state of near-maximal homeostatic (automatic), chemoreflex drive induced by the introduction of hypoxic and/or hypercapnic gas mixtures (10% $O_2$, 5% $CO_2$ or 21% $O_2$, 10% $CO_2$) to the plethysmograph chamber. Then, room air was restored, and 1.5% isoflurane was introduced to assess breathing patterns in the lightly anesthetized, unresponsive, state. Example plethysmograph recordings with corresponding instantaneous breathing frequency and acceleration (ABS (Hz/s)) are shown for each condition in Fig. 1b$_{1-4}$. Peak inspiratory pressure, inspiratory duration ($T_I$), expiratory duration ($T_E$), instantaneous frequency, and area were quantified (Fig. 1a), and breaths under spontaneous conditions ($n = 22,345$, Fig. 1c) or across all conditions ($n = 67,579$; Fig. 1d and Fig. S1) were plotted in principle-component space. Breaths during rapid breathing bouts formed a cluster that was distinct from other breathing patterns under spontaneous conditions, breathing patterns driven by chemoreflexes, or under isoflurane anesthesia. These rapid breathing patterns generally occurred as abrupt, high acceleration, transitions from slow, regular breathing and occurred in bouts (Fig. 1b, e). Indeed, in the awake state under control conditions, mice exhibited a wide range of breathing frequencies, which was bimodally distributed with a slower mode near 3–4 Hz and more rapid mode near 9-10 Hz reaching a maximum of ~13 Hz. Under conditions of near-maximal chemoreflex drive, breathing frequencies became more narrowly and unimodally distributed around 5–6 Hz, matching frequencies that have been previously achieved via optogenetic manipulations of rhythmogenic preBötC neurons[36,38]. Breathing also became much less dynamic, as measured by respiratory acceleration, with very few bouts of rapid breathing exceeding 8 Hz (Fig. 1b, e, f). Upon induction of light anesthesia, breathing became highly regular and all high-frequency respiratory activity was lost, leaving a unimodal distribution of frequency with a peak near 3–4 Hz, aligned with the slower mode of frequencies observed under awake control conditions. These results are consistent with prior observations in rodents[10,12,53,54] and emphasize that rapid and dynamic breathing patterns, likely reflecting behavioral and emotional modulation[14,15,18,55], are unique to the awake state and are suppressed by automatic respiratory control reflexes when physiological demands are increased.

### Peptidergic neuron populations in the PBN exert opposing and conditional effects on rapid breathing
Manipulations of the preBötC by us[38,56] and others[36,37,57–59] have been unable to produce rapid breathing frequencies comparable to those common in awake spontaneously behaving mice (see Fig. 1f). Thus, to begin to elucidate the neurocircuitry that may underlie these rapid breathing patterns, we turned to the parabrachial nucleus (PBN) in the dorsolateral pons, a region that is not considered essential for respiratory rhythmogenesis per se but is an important source of modulatory input[20,22,23]. Non-cell type specific electrical or chemical stimulations of the PBN elicit varied effects on breathing, capable of producing either increases or decreases in respiratory activity[60,61], suggesting there are multiple subpopulations of PBN neurons with distinct respiratory functions. Yet, the respiratory roles of specific transcriptionally defined PBN neuronal subtypes are not well defined compared to those of the preBötC, and targeted manipulations have primarily involved neurons that express the mu opioid receptor (MOR), encoded by *Oprm1*[17,51], which is widely expressed across many PBN neurons in its lateral, medial, and Kölliker-Fuse (KF) subdivisions[48,62,63].

To assess the potential respiratory role of more restricted groups of PBN neurons, we investigated neurons that express Tac1 (precursor of Substance P and other tackyinins) or CGRP. These peptides are strongly expressed in subsets of neurons primarily localized to the lateral PBN. To determine the degree of overlap between these peptidergic subpopulations and their relationship to MOR neurons, we performed RNAScope to label cells expressing *Oprm1*, *Tac1*, and *Calca*

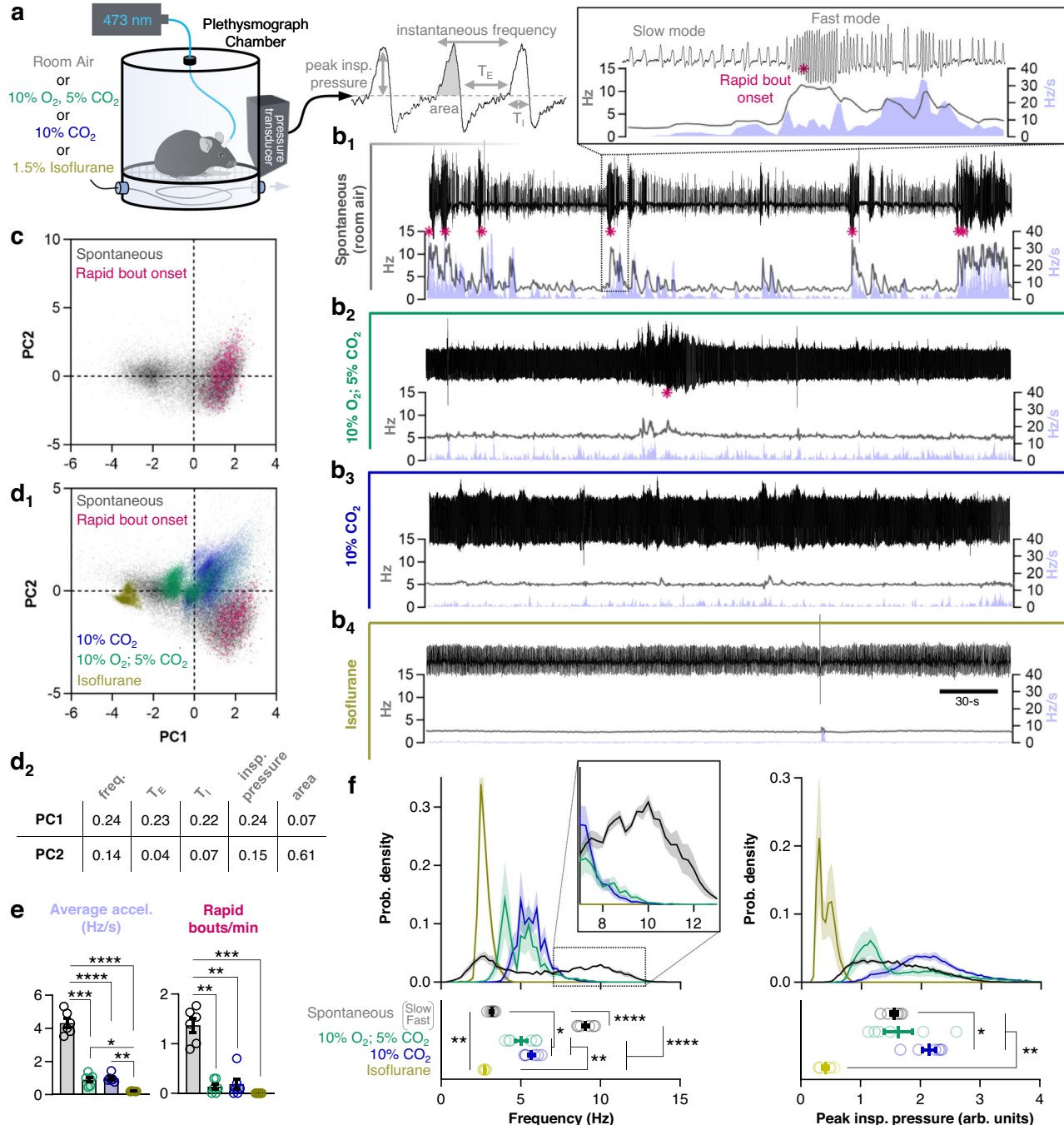

**Fig. 1 | Rapid dynamic breathing patterns characterize the awake, normally behaving, state. a** Schematic of experimental setup and example plethysmograph recording showing quantified parameters. **b** Representative recordings from an awake freely behaving mouse breathing room air ($b_1$), during near maximal chemoreflex stimulation with 10% $O_2$; 5% $CO_2$ ($b_2$) or 10% $CO_2$ ($b_3$) and following transition to the anesthetized state with 1.5% isoflurane in room air ($b_4$). Panels show plethysmograph pressure waveforms (top) and corresponding 3-breath moving average of breathing frequency (black) and breathing acceleration (violet). Magenta stars indicate breathing bouts that exceed 8 Hz. **c** PCA of breathing patterns generated spontaneously in the awake state (22,345 breaths from $n = 6$ mice). Magenta colored dots correspond to the onset (first 5 breaths) of rapid breathing bouts as shown in $b_1$. **d** PCA of spontaneous, chemoreflex-driven, and anesthetized

breathing patterns ($d_1$; 67,579 breaths from $n = 6$ mice) and relative contributions of respiratory parameters to PCs 1 and 2 ($D_2$). **e** Quantified average acceleration (left) and number of rapid breathing bouts per minute (right). $n = 6$; one-way RM ANOVA with Tukey's multiple comparisons tests. **f** Probability density histograms (top) and mean values from each mouse ($n = 6$; bottom) comparing breathing frequencies (left) and peak inspiratory pressures (right) during each condition. Inset illustrates the rapid breathing frequencies (~8–12 Hz) that characterize the awake spontaneously breathing state. Breathing frequencies under spontaneous conditions were bimodal and were separated into "slow" and "fast" modes for analysis. One-way RM ANOVA with Tukey's multiple comparisons tests. *$p < 0.05$; **$p < 0.01$; ***$p < 0.001$; ****$p < 0.0001$. Means±SE. Source data are provided as a Source Data file.

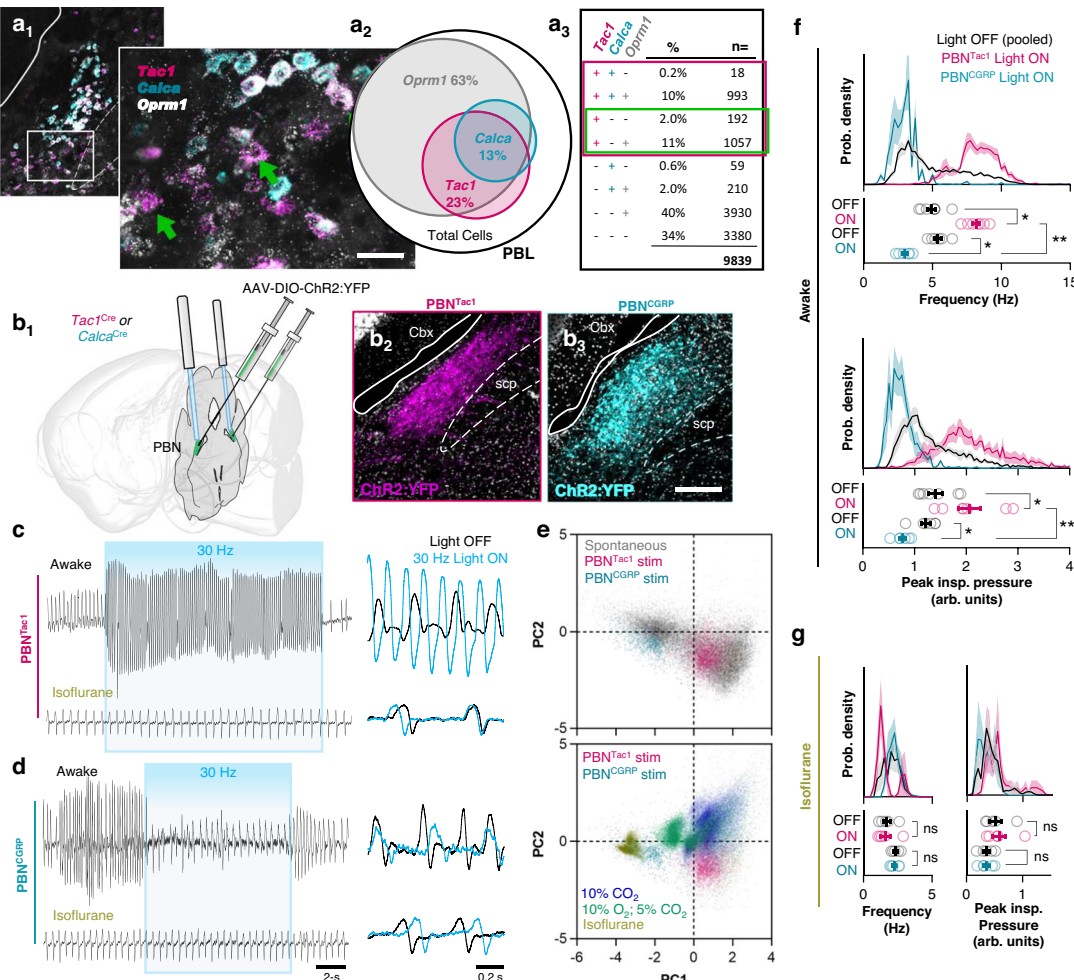

**Fig. 2 | Peptidergic PBN subpopulations have opposing effects on breathing.**
**a** Example RNAscope images illustrating partially overlapping *Tac1*, *Calca*, and *Oprm1* populations in the PBL (a₁) with Venn Diagram (a₂) and table (a₃) showing the estimated size of each subpopulation as a fraction of total PBL cells (*n* = 9839 cells analyzed from *n* = 3 mice). Green arrows in a₁ and green box in a₃ indicate *Tac1* neurons that do not co-express *Calca*. Scale bars = 40 μm and 20 μm (zoomed). **b** Schematic of viral gene transfer approach (b₁) and recombinase-dependent expression of ChR2:YFP in Tac1 (b₂) or CGRP (b₃) PBN neurons representative of *n* = 7 (Tac1) or *n* = 6 (CGRP) mice. Scale bar = 200 μm. scp superior cerebellar peduncle, Cbx cerebellar cortex. **c** Representative plethysmograph recordings from a mouse when awake (top) and lightly anesthetized with 1.5% isoflurane (bottom) during photoactivation of PBN^Tac1 neurons for 15 s at 30 Hz. Panels on right show overlay of 1 s of breathing with light OFF (black) and light ON (blue).

**d** Same as **c** but during photoactivation of PBN^CGRP neurons. Scale bars in **c** also apply to **d**. **e** PCA comparing spontaneous breathing patterns (22,351 breaths from *n* = 6 mice) to breathing patterns during photoactivation of PBN^Tac1 (3,507 breaths from *n* = 7 mice) and PBN^CGRP (848 breaths from *n* = 6 mice) neurons. **f** Probability density histograms (top) and mean values from each mouse (bottom; *n* = 7 Tac1, *n* = 6 CGRP) comparing breathing frequencies and peak inspiratory pressures during photoactivation of Tac1 or CGRP PBN neurons. Tac1 and CGRP data were pooled for the light OFF condition shown in the probability density histograms. Light OFF and ON conditions and Tac1 vs. CGRP Light ON conditions were compared using two-tailed Wilcoxon matched pairs and Mann–Whitney tests, respectively. **g** Same as **f** but in the isoflurane anesthetized state (abscissa scaled to match **f**). *$p < 0.05$; **$p < 0.01$. Means±SE. Source data and statistical details provided in Source Data file.

mRNA followed by automated cell counting (see Methods) (Fig. 2a₁₋₃). *Oprm1* was widely expressed, with 62% of the *n* = 9839 cells analyzed expressing detectable levels of *Oprm1* mRNA, whereas 23% and 13% of total cells expressed *Tac1* and *Calca* mRNA, respectively. Most *Tac1* (91%) and *Calca* (94%) neurons also expressed *Oprm1*; however, the majority of *Oprm1* neurons (63%) did not express mRNA for either peptide. There was also considerable overlap between *Tac1* and *Calca* neuronal subpopulations, with 45% and 79% of *Tac1* and *Calca* neurons co-expressing mRNA for both peptides, respectively.

To examine the roles of these peptidergic subpopulations for breathing, *Tac1^Cre/+* or *Calca^Cre/+* mice received bilateral injections of AAV-DIO-ChR2:YFP into the PBN to allow specific activation of either Tac1 or CGRP neurons with light via optical fibers implanted above the PBN (Fig. 2b₁₋₃). Changes in breathing activity elicited during photostimulation were then assessed under control conditions (room air) in the awake state and again following induction of anesthesia with 1.5%

isoflurane. During photoactivation (10-ms pulses at 30 Hz for 15 s) of Tac1 neurons in the PBN, an immediate and sustained change in breathing pattern was elicited (Fig. 2c and Fig. S2a) that overlapped in principle-component space with breaths produced during spontaneous rapid bouts but was distinct from chemoreflex driven breathing patterns (Fig. 2e). Indeed, photostimulation of Tac1 PBN neurons led to a large increase in breathing frequency to $8.1 \pm 0.3$ Hz (Fig. 2f), which exceeded frequencies produced during near-maximal chemoreflex activation ($5.7 \pm 0.2$ Hz in 10% $CO_2$) and was commensurate with the rapid mode of frequencies generated spontaneously under control conditions ($9.1 \pm 0.2$ Hz) (Fig. S2b). Tac1 neuron photostimulation also increased peak inspiratory pressure ($45 \pm 5$%; Fig. 2f and Fig. S2a), such that breath waveforms were similar to those produced during spontaneous rapid breathing bouts (Fig. S2c). In contrast, activation of CGRP neurons elicited a respiratory pattern that resembled resting breathing while awake or lightly anesthetized (Fig. 2e, and Fig. S2c).

Specifically, photostimulation terminated any ongoing high-frequency breathing patterns and caused a modest suppression of breathing frequency to $3.0 \pm 0.5$ Hz[64] and peak inspiratory pressure ($23 \pm 6\%$) (Fig. 2d and Fig. S2a). Thus, despite Tac1 and CGRP being co-expressed in many neurons of the lateral PBN (see above and[48]), their activation elicits opposing effects on breathing in the awake state, with Tac1 neurons promoting rapid breathing patterns and CGRP neurons suppressing them.

To assess the role of these PBN subpopulations in the regulation of automatic respiratory control, photostimulations were repeated following induction of light isoflurane anesthesia. Unexpectedly, breathing patterns in the anesthetized state were unaffected by activation of either Tac1 or CGRP neurons (Fig. 2c, d, g and Fig. S2d). This finding is intriguing because it contrasts with the effects of preBötC manipulations, which generally produce similar or even more robust changes to the respiratory rhythm in the anesthetized versus awake state[36,38], suggesting that these peptidergic subpopulations exert conditional control of breathing linked to its state-dependent regulation.

To further test the state-dependent roles of these peptidergic subpopulations, we compared the effects of Tac1 or CGRP neuron stimulation to stimulation of the broader MOR-expressing population of PBN neurons under the same experimental conditions. AAV-DIO-ChR2:YFP was injected into the PBN of $Oprm1^{Cre/+}$ mice. MOR neurons were then activated with light via implanted optical fibers and changes in breathing activity were assessed under control conditions (room air) in the awake state and again following induction of isoflurane anesthesia. Consistent with the opposing functional roles of Tac1+ and CGRP+ subpopulations (see Fig. 2), the effects were complex. In 5 of 8 mice, breathing was increased, whereas it was suppressed in 3 of 8 (Max: 8.6 Hz, Min: 0 Hz; Fig. S3a–c), possibly reflecting the broader topographic localization of MOR neurons across PBN subregions[48,60]. Importantly, effects on breathing persisted following induction of anesthesia, but became more modest than in the awake state (Fig. S3b–d), perhaps due to loss of the state-dependent contributions of Tac1 and CGRP neurons. These results support the idea that the PBN contains multiple subpopulations of MOR expressing neurons that differentially contribute to both the automatic and state-dependent control of breathing.

### PBN neurons that express Tac1 but not CGRP exert potent control of breathing that is state-dependent and respiratory phase-independent

To further dissect the roles of Tac1 and CGRP neuronal subpopulations in the PBN (see Fig. 2a), we utilized an intersectional approach combining dual-transgenic recombinase-expressing mice with INTRSECT molecular targeting[65,66]. To do so, a $Calca^{FLPo}$ mouse line was generated in house (see Methods; Fig. S4) and bred to $Tac1^{Cre}$ mice. Offspring heterozygous for both transgenes ($Tac1^{Cre/+}$; $Calca^{FLPo/+}$) received bilateral injections of a dual-recombinase-dependent AAV vector with $Cre_{ON}$ $FLPo_{OFF}$ logic to express ChR2:YFP specifically in PBN neurons that express Tac1, but not CGRP (Fig. 3a). Like activation of all Tac1 neurons, photostimulations (10-ms pulses at 30 Hz for 15 s) of this CGRP-negative subset of Tac1 neurons (~13% of total PBL cells), referred to here as Tac1+; CGRP- neurons, produced a potent and sustained shift in breathing to a respiratory pattern that overlapped in principle-component space with spontaneous rapid breathing bouts, but was distinct from chemoreflex-driven breathing (Fig. 3b, c and Fig. S5a–c). Remarkably, even higher frequencies were achieved by more specific activation of Tac1+; CGRP- neurons ($10.4 \pm 0.3$ Hz; Fig. 3d), consistent with Tac1 and CGRP PBN subpopulations having opposing effects on breathing in the awake state. Similar results were observed during 10- and 20-Hz photostimulations (Fig. S6) and breathing patterns during light OFF and light ON conditions

remained consistent over the course of repeated trials (Fig. S7). To examine the potential state-dependence of respiratory control by Tac1+; CGRP- neurons, 30-Hz photostimulations were repeated following induction of anesthesia with either 1.5% isoflurane or 1.5 g/kg urethane (Fig. 3b), an anesthetic commonly used to study the control of breathing due to its potentially more modest effects on cardiorespiratory function[40,67,68]. Consistent with our observations during stimulations targeting all Tac1 neurons, changes in respiratory frequency and peak inspiratory pressure evoked by stimulation of Tac1+; CGRP- PBN neurons were eliminated following induction of either isoflurane or urethane anesthesia (Fig. 3b, e, f and Fig. S5b), suggesting that this subgroup of Tac1-expressing PBN neurons exerts profound, state-dependent regulation of breathing with minimal effects on its underlying automatic control.

To further examine how activation of Tac1+; CGRP- neurons affects the ongoing breathing rhythm, we conducted respiratory-phase-resetting experiments. Brief 25-ms light pulses, separated by >30 s, were delivered such that they occurred at random time-points within an ongoing respiratory cycle (defined as start of inspiration $n$ to start of inspiration $n + 1$) (Fig. 3g). The lengths of respiratory cycles containing light pulses were compared to cycle lengths during sham stimulations where no light was delivered, and cycle lengths of all individual trials were normalized to the mean cycle length during sham stimulations (Fig. 3h, i). Importantly, in the awake state, we found that brief activation of Tac1+; CGRP- neurons induced a strong respiratory-phase advance, independent of when the light pulse occurred during the ongoing cycle (Fig. S5d). Indeed, regardless of whether light pulses occurred during inspiration or expiration, respiratory cycle length was shortened to $23.3 \pm 2.4\%$ and $28.7 \pm 2.8\%$ of sham stimulations, respectively (Fig. 3i and Fig. S5d, e) with an average latency of $42.0 \pm 1.1$ ms from stimulation to the onset of the next breath ($n = 277$ stimulations). The respiratory-phase independent effects of Tac1+; CGRP- neurons on the breathing rhythm contrast with manipulations of the preBötC, which can either lengthen or shorten the respiratory cycle depending on whether the stimulus occurred during inspiration or expiration[36,56,57]. When the same mice were anesthetized with isoflurane or urethane, light pulses no longer evoked changes in respiratory cycle length (Fig. 3g–i), further demonstrating the striking state-dependence of the respiratory control exerted by Tac1+; CGRP-neurons.

### Tac1+; CGRP- PBN neurons exert precise temporal control of breathing throughout the physiological range

In addition to the respiratory phase-resetting experiments described above, the ability of a stimulus to entrain the frequency of an oscillation implies that the activated neurons either participate in or strongly influence the underlying rhythm-generating circuitry[2,22,69]. To examine the extent to which Tac1+; CGRP- neurons in the PBN can entrain respiratory rhythmogenesis, awake mice were subjected to 15-s trains of 20-ms light pulses delivered at frequencies ranging from 3 Hz to 15 Hz, exceeding the maximum rates observed in spontaneously breathing mice (see Fig. 1). Example plethysmograph recordings during 6-, 8-, 10-, and 12-Hz stimulation of Tac1+; CGRP- neurons are shown in Fig. 4a$_{1-3}$. At slower stimulation frequencies (<7 Hz), breathing rate was often increased such that it outpaced the stimulus (Fig. 4b, e and Fig. S8a). Indeed, even single 20-ms light pulses often induced an increase in breathing frequency that outlasted the initial respiratory phase advance (see Fig. 3g and Fig. S8b), indicative of hysteresis in the underlying mechanisms that mediate the changes in breathing elicited by Tac1+; CGRP- neurons. At higher stimulation frequencies (>7 Hz), breathing rate became entrained (Fig. 4a, b, d) such that one breath was elicited per light pulse (Fig. 5e) with increasing cross-correlation between the plethysmograph and laser-stimulus waveforms (Fig. 4c). Lag times producing peak cross-correlations were between 40-60 ms (Fig. 4g), consistent with the above respiratory-phase resetting

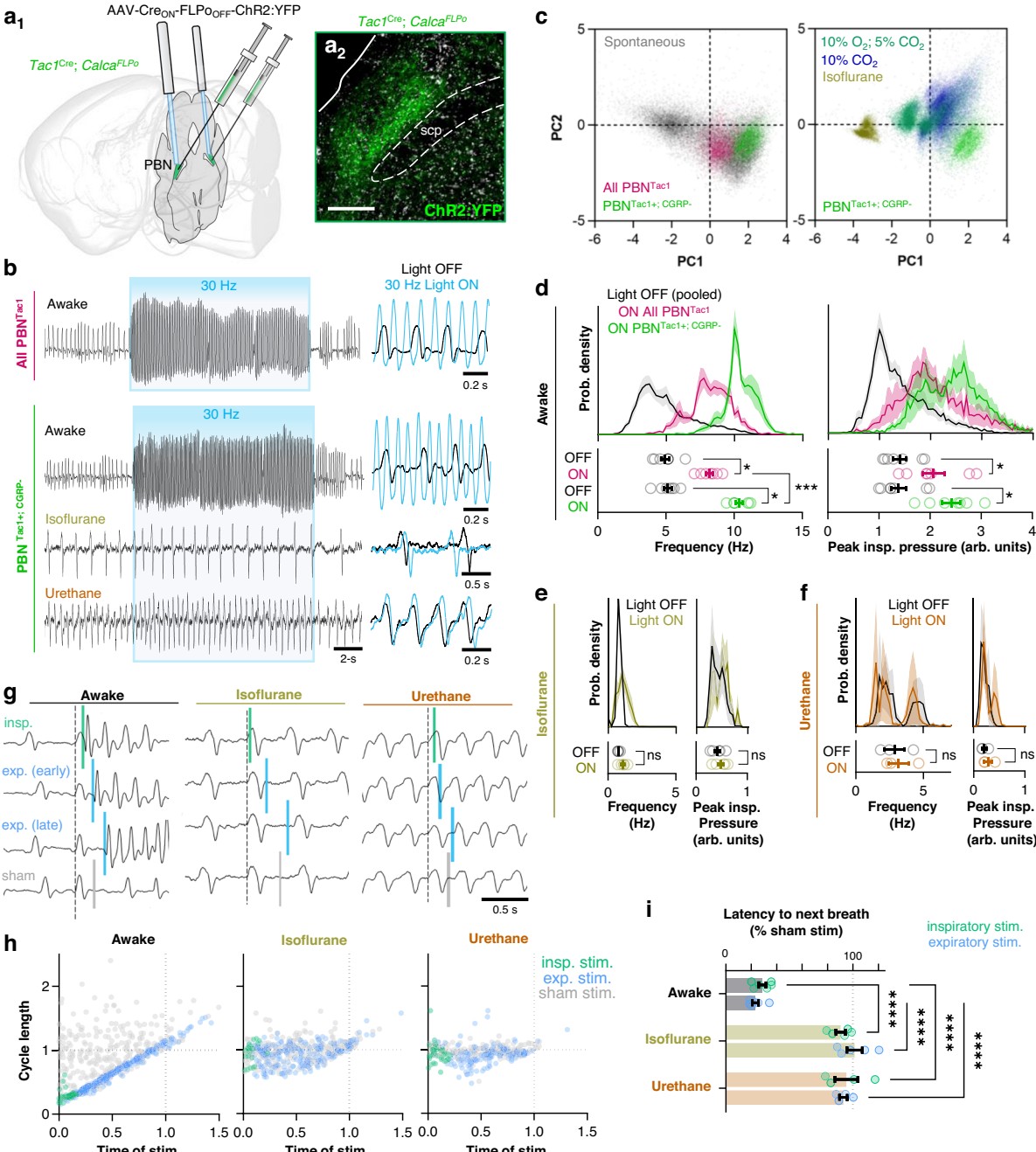

**Fig. 3 | PBN neurons that express Tac1 but not CGRP exert potent control of breathing that is state-dependent and respiratory phase-independent. a** Dual recombinase-dependent gene transfer approach to direct ChR2:YFP to neurons that express Tac1 but not CGRP ($a_1$) and representative histology of viral expression ($a_2$) in $n = 7$ mice. Scale bar = 200 μm. **b** Plethysmography recordings comparing 30-Hz photostimulations (15 s) when all Tac1 neurons are targeted versus targeting of Tac1 neurons that do not co-express CGRP. Photostimulations were repeated following induction of isoflurane or urethane anesthesia. Overlays of breathing patterns during 1 s of light OFF (black) and light ON (blue) conditions shown on right. **c** PCA comparing spontaneous breathing patterns (22,351 breaths, $n = 6$ mice) to breathing patterns during photoactivation of $PBN^{Tac1}$ (3,507 breaths, $n = 7$ mice) and $PBN^{Tac1+; CGRP-}$ (3755 breaths, $n = 7$ mice) neurons. **d** Probability-densities (top) and mean values from each mouse (bottom; $n = 7$ Tac1, $n = 7$ Tac1+; CGRP-) comparing breathing frequencies (left) and peak inspiratory pressures (right) during photostimulations. Tac1 and Tac1+; CGRP- data were pooled for the light OFF condition

shown in the probability density histograms. Light OFF versus ON conditions and Tac1 versus CGRP Light ON conditions were compared using two-tailed Wilcoxon matched pairs and Mann–Whitney tests, respectively. **e** and **f** Same as **d** but in the isoflurane or urethane anesthetized state. **g** Breathing responses to 25-ms light pulses or sham stimulations delivered at different phases of the respiratory cycle in the awake or anesthetized state. **h** Plots showing the effects of brief photoactivations of $PBN^{Tac1+; CGRP-}$ neurons on respiratory cycle length relative to the time of stimulation (abscissa and ordinate normalized to mean sham respiratory cycle length). Green and blue dots represent stimulations ($n = 283$ awake, $n = 223$ isoflurane, $n = 160$ urethane) occurring during inspiration or expiration, respectively; sham stimulations shown in light gray. **i** Latency from photostimulation to the onset of the next breath expressed as a % of sham stimulations during inspiration or expiration. Mixed effects model with Tukey's multiple comparisons tests ($n = 6$ awake, $n = 5$ isoflurane, $n = 4$ urethane). *$p < 0.05$; ***$p < 0.001$; ****$p < 0.0001$. Means ± SE. Source data and statistical details provided in Source Data file.

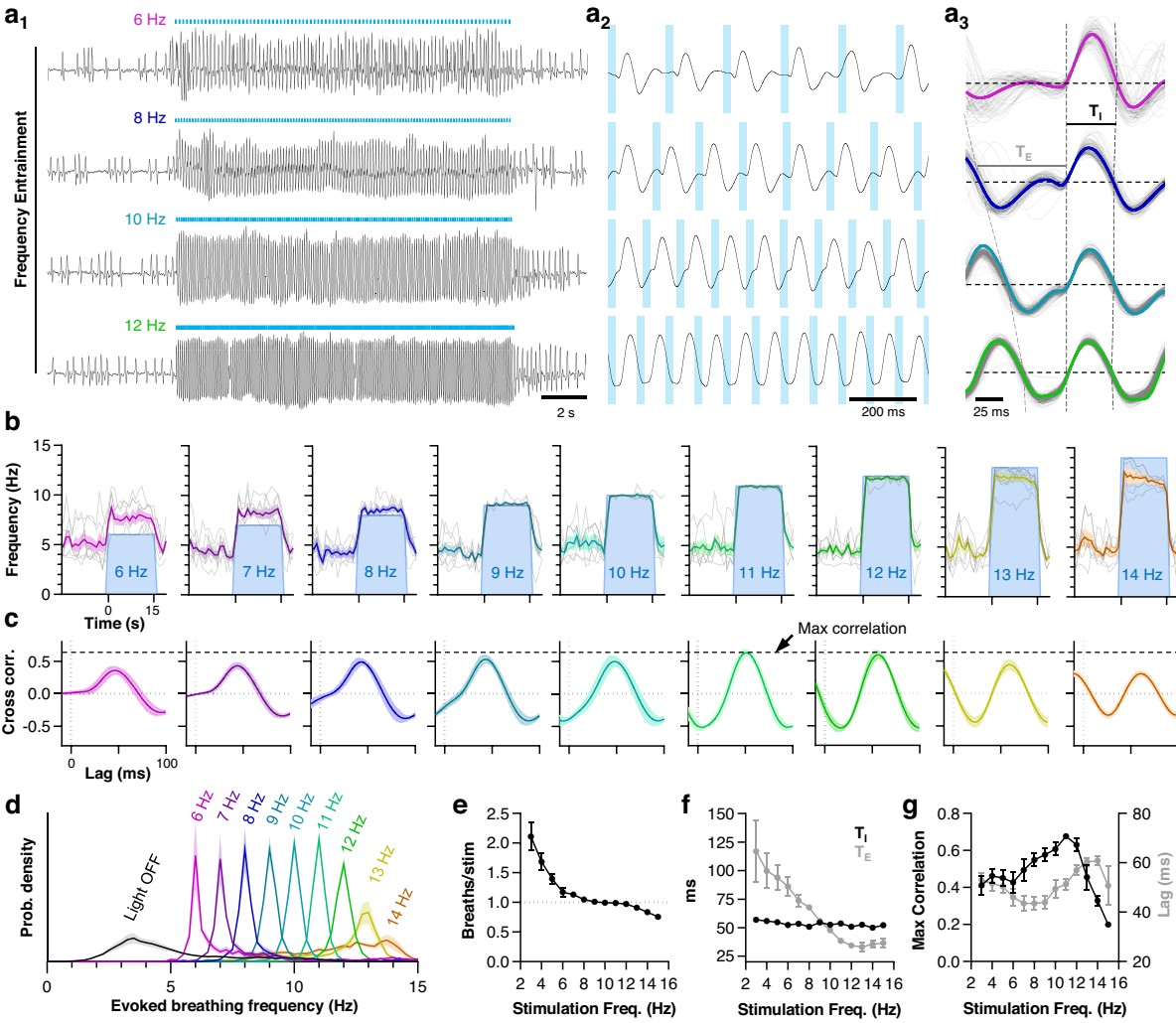

**Fig. 4 | Tac1+; CGRP- PBN neurons exert precise temporal control of breathing throughout the maximum physiological range. a** Example plethysmograph recordings showing entrainment of the respiratory rhythm during 15-s photostimulations of Tac1+; CGRP- neurons at 6, 8, 10, or 12 Hz (a₁). Traces are expanded in a₂, and overlaid and averaged (bolded lines) in a₃ to illustrate relative effects on $T_I$ and $T_E$. **b** Average breathing frequency (replicates shown in light gray) before and during 15-s photostimulations of Tac1+; CGRP- PBN neurons at 6–14 Hz ($n = 7$ mice). **c** Cross-correlations of laser pulse and plethysmograph waveforms for data shown in **b**. Note optimal correlation and entrainment at 11 Hz stimulation frequency.

**d** Probability density histograms of evoked breath frequencies during 6–14 Hz photostimulations relative to the light OFF condition (data pooled from all stimulation frequencies). **e** Average number of breaths per stimulation for each mouse ($n = 7$) relative to the stimulation frequency showing a near 1:1 relationship at 8–12 Hz. **f** Comparison of $T_I$ and $T_E$ of the evoked respiratory rhythm, and **g** comparison of maximum cross correlation and corresponding lag time relative to stimulation frequency ($n = 7$). Means ± SE. Source data and statistical details provided in Source Data file.

experiments which revealed an average latency of 42 ± 1 ms from light onset to the onset of the next inspiration. Strikingly, entrainment improved (indicated by peak cross-correlation) as the stimulus frequency was increased until reaching an optimum at 11 Hz, and breathing remained consistently entrained throughout the entire 15-s stimulus period at frequencies up to 12 Hz (13 Hz in some mice) — breathing rates that are commensurate with the physiological maximum for mice (Fig. 1 and refs. [13,14]). Incremental increases in entrained breathing rate were achieved through shortening of the time between inspirations (expiratory time, $T_E$) without similar changes in the duration of inspiration ($T_I$) (Fig. 4a₃, f). At breathing frequencies near the physiological maximum, $T_E$ reached a minimum and plateaued. At stimulation frequencies that exceed this physiological limit (≥13 Hz), breathing became more irregular, correlations became weaker, and breathing rate could not be further increased. These results demonstrate the unprecedented ability of Tac1+; CGRP- neurons to precisely control rapid breathing in awake mice.

**The state-dependent control of breathing by Tac1+; CGRP- PBN neurons is mediated by direct projections to the ventral medulla**
In addition to gene expression profiles, distinguishing neurons based on their network interactions and axon projections is a powerful strategy for defining functional circuits[70,71]. For the PBN, manipulating specific projections of transcriptionally defined subpopulations[45,46,48,72] has been a critical step in unraveling their diverse functional roles. Thus, we asked which projections of Tac1+; CGRP- PBN neurons mediate their potent control of breathing. While projections to the preBötC in the ventral medulla were an obvious candidate due to the known role of this region in respiratory rhythm generation, we anticipated that projections to forebrain region(s) linked to behavioral and emotional regulation would be equally plausible given the striking state-dependence of respiratory control by Tac1+; CGRP- neurons. Indeed, previous work has found that non-specific electrical stimulation of the central nucleus of the amygdala (CeA), which has direct connections with the preBötC[32], could entrain breathing in awake, but

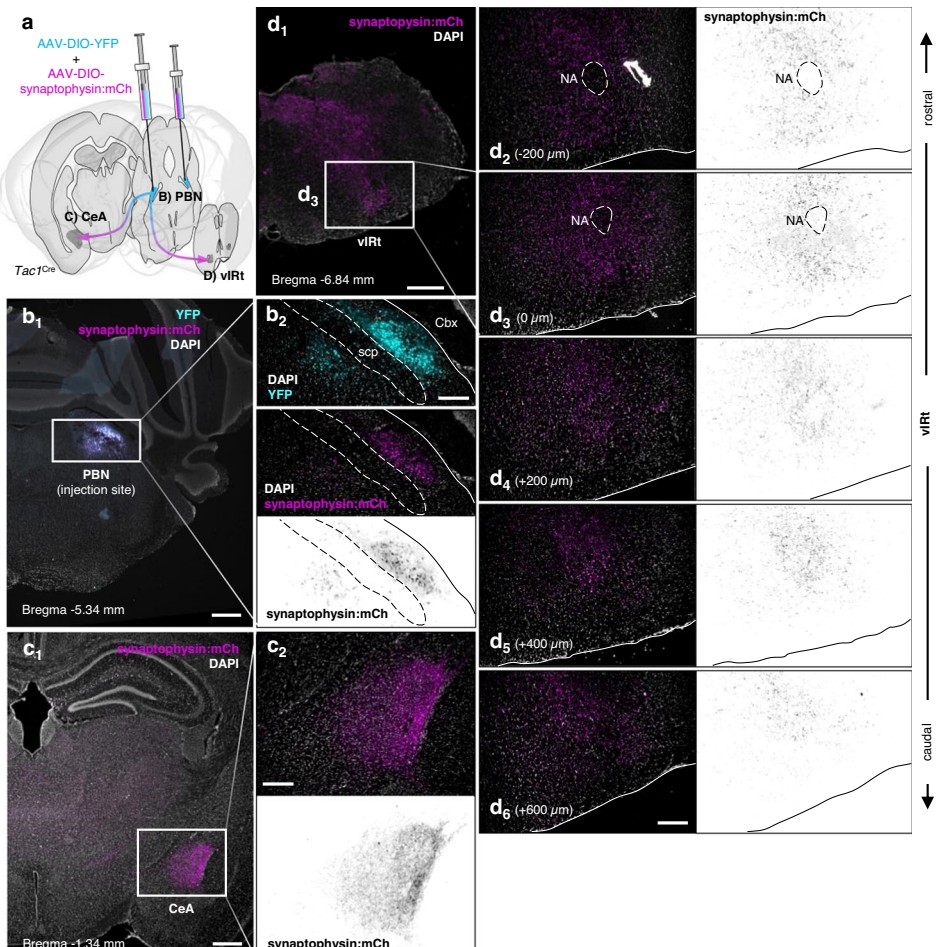

**Fig. 5 | Tac1 PBN projections to the CeA and vlRt. a** Schematic of experimental approach using co-injection of AAV-DIO-YFP and AAV-DIO-synaptophysin:mCh to label PBN^Tac1 cell bodies and synapses, respectively. **b** Labeling of Tac1 neurons (YFP pseudocolored cyan) in the PBN and their synapses (synaptophysin:mCh pseudo-colored magenta). **c** Labeling of Tac1 synapses in the CeA. **d** Labeling of

Tac1 synapses in the vlRt. Scale bars = 500 μm for b$_1$, c$_1$, and d$_1$. Panels to the right (b$_2$, c$_2$, d$_{2-6}$) show 10X magnification and inverted grayscale images of synapto-physin:mCh for enhanced contrast. Scale bars = 200 μm. Sections in d$_{2-6}$ are arranged rostral to caudal corresponding to ~−6.7 mm to −7.5 mm relative to Bregma. NA nucleus ambiguus. Images representative of $n$ = 3 replicates.

not sleeping, cats[73]. Therefore, we chose to compare the functional role of Tac1+; CGRP- projections from the PBN to the CeA and vlRt of the medulla, encompassing the preBötC.

An initial histological analysis of the brains of mice that received injections of INTRSECT AAV to label Tac1+; CGRP- neurons revealed a projection pattern similar to what has been described for other PBN populations, including PBN^CGRP neurons, with fibers present in the lateral septal nucleus, bed nucleus of the stria terminalis, CeA, ven-troposteromedial nucleus of the thalamus, and ventral medulla. Because observed fibers could be fibers of passage, in a follow-up experiment we co-injected two AAVs into the PBN of *Tac1*^Cre/+ mice, one carrying a Cre-dependent reporter to fill cell bodies (AAV-DIO-YFP), and another encoding a reporter that is preferentially localized to synapses (AAV-DIO-synaptophysin:mCherry) (Fig. 5). Synaptophy-sin:mCherry labeling was observed in the CeA (Fig. 5c) and extended rostrocaudally along the vlRt (Fig. 5d), confirming both regions as plausible downstream targets of PBN neurons that could mediate their control of breathing.

To manipulate the activity of Tac1+; CGRP- PBN projections to the CeA or vlRt, *Tac1*^Cre/+; *Calca*^FLPo/+ mice were injected with AAV carrying a Cre$_{ON}$ FLPo$_{OFF}$ vector expressing ChR2:YFP. Mice were then implanted with optical fibers positioned bilaterally over terminal fields in the vlRt and the CeA (Fig. 6a; Fig. S9). Like stimulation of Tac1+; CGRP- cell bodies, photostimulation (10-ms pulses at 30 Hz for 15 s) of ChR2-

expressing fibers in the vlRt produced a respiratory pattern that over-lapped in principle-component space with breaths produced during rapid breathing bouts in normally behaving awake mice (Fig. 6b, c). Accordingly, stimulation of Tac1+; CGRP- terminals in the vlRt elicited a consistent and sustained increase in breathing frequency (8.1 ± 0.3 Hz) and peak inspiratory pressure (43.1 ± 6.7%; Fig. 6d and Fig. S10a–c). In contrast, 15-s stimulations of Tac1+; CGRP- terminals in the CeA had inconsistent effects (Fig. 6b), with breathing rates remaining widely distributed, and induced only a modest increase in frequency (6.4 ± 0.6 Hz) and peak inspiratory pressures (15.9 ± 3.7%; Fig. 6c, d and Fig. S10a–c). To further compare the effects of Tac1+; CGRP- projections to the vlRt and CeA on breathing, we performed phase-resetting experiments by delivering 25-ms light pulses randomly during the respiratory cycle (Fig. 6f). Consistent with stimulation of Tac1+; CGRP-cell bodies in the PBN, photostimulation of their descending projec-tions to the vlRt caused a potent shortening of respiratory cycle length regardless of whether the stimulation occurred during inspiration or expiration (28.8 ± 3.3% and 41.4 ± 4.1% of sham cycle durations, respectively; Fig. 6f–h) with an average latency of 54.9 ± 1.3 ms ($n$ = 240 stimulations) from stimulus onset to the subsequent breath. On the other hand, brief stimulations of Tac1+; CGRP- neuronal pro-jections to the CeA elicited no changes in respiratory cycle length (86.7 ± 2.7% and 101.5 ± 12.5% of sham stimulations during inspiration and expiration, respectively) with an average latency of 149.1 ± 8.1 ms

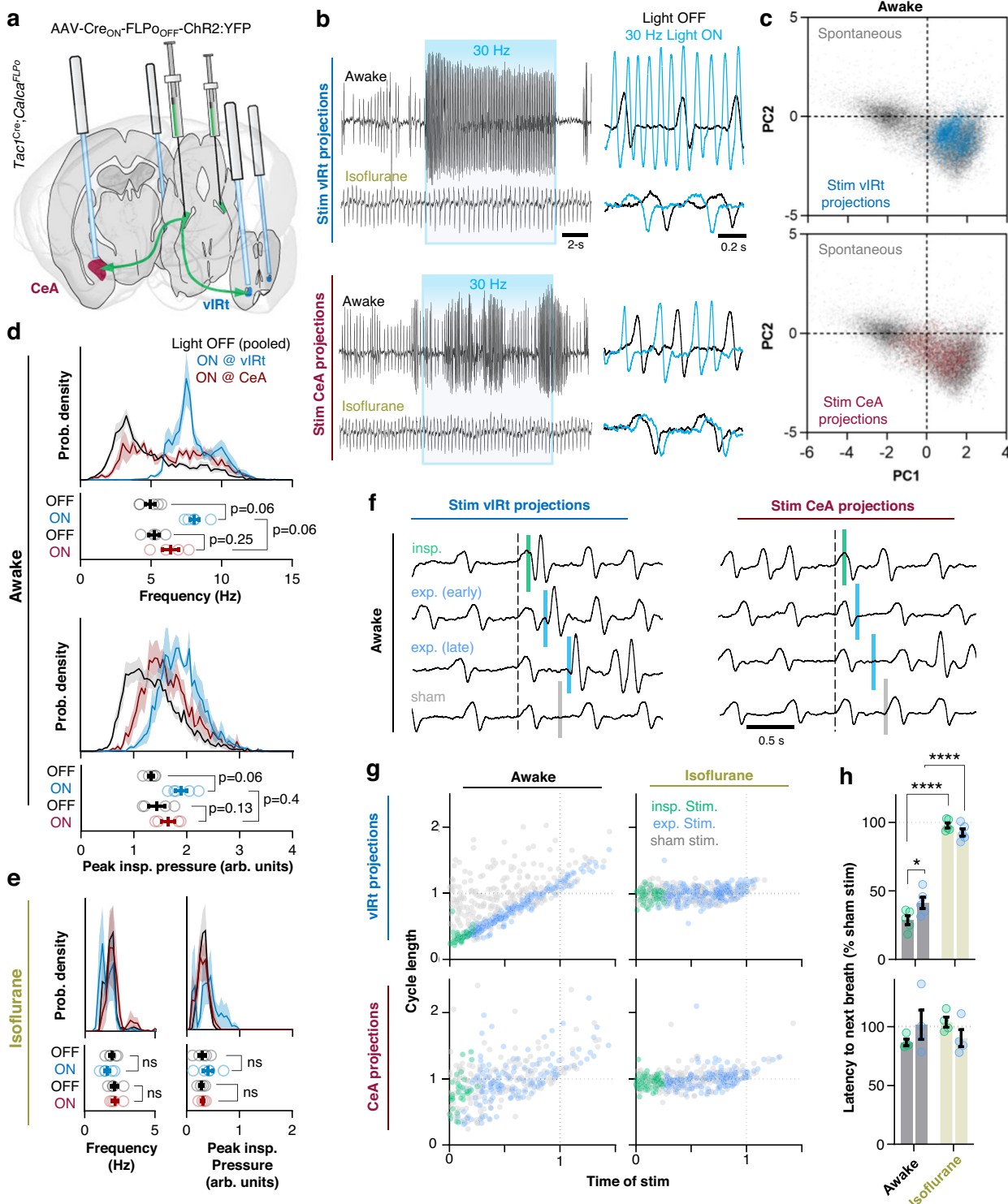

from the stimulation to the next breath (Fig. 6f–h and Fig. S10d). Upon induction of light isoflurane anesthesia, the effects of activation of Tac1+; CGRP- neuronal terminals in the vIRt were eliminated (Fig. 6b, d–g and Fig. S10b). In some mice, the ventral surface of the medulla was surgically exposed under urethane anesthesia to allow stronger photostimulations of PBN projections in the vIRt while directly recording respiratory motor output from the hypoglossal (XII) nerve (Fig. S11, S12). No effects of breathing were elicited during continuous bilateral activation of Tac1+ PBN projections with 200-μm diameter fibers at 2 mW (power used in the awake state), 4 mW, or 6 mW (Fig. S11). Similarly, strong stimulations (30-Hz at 6 mW,) of Tac1+; CGRP- projections in the

vIRt also failed to elicit changes in breathing pattern or induce respiratory phase-shits (Fig. S12). Together, these results indicate that Tac1+; CGRP- neurons in the PBN exert potent state-dependent and respiratory phase-independent control of breathing primarily via direct descending projections to the medullary vIRt.

## Discussion

Extensive research has focused on unraveling the neural circuits that control breathing, and much is known regarding the brainstem, spinal, and peripheral mechanisms that automatically produce respiratory rhythmic activity, adjust breathing to changes in chemo- and

**Fig. 6 | State-dependent breathing control by Tac1+; CGRP- PBN neurons is mediated by direct projections to the ventral medulla. a** Approach to activate axon terminals of Tac1+; CGRP- PBN neurons in the CeA or vIRt. **b** Plethymograph recordings comparing 30-Hz photostimulations (15 s) of Tac1+; CGRP- terminals in the vIRt (top) or CeA (bottom) in the awake or anesthetized state. Panels on right show overlayed breathing patterns during 1 s of light OFF (black) and light ON (blue) conditions. **c** PCA comparing spontaneous breathing patterns (22,351 breaths, $n = 6$ mice) to breathing patterns during photoactivation of Tac1+; CGRP- projections to the vIRt (top; 3330 breaths, $n = 5$ mice) or CeA (bottom; 1966 breaths, $n = 4$ mice). **d** Probability densities and mean values from each mouse ($n = 5$ vIRt, $n = 4$ CeA) comparing breathing frequencies (top) and peak inspiratory pressures (bottom) during 30-Hz photostimulations. vIRt and CeA data were pooled for the light OFF condition shown in the probability density histograms. Light OFF versus ON conditions and vIRt versus CeA Light ON conditions were compared using two-tailed Wilcoxon matched pairs and Mann–Whitney tests,

respectively. **e** Same as **d** but following induction of isoflurane anesthesia. **f** Plethysmograph recordings showing breathing responses to 25-ms photo-stimulations (or sham) of Tac1+; CGRP- projections in the vIRt (left) or CeA (right) at different phases of the respiratory cycle in an awake mouse. **g** Quantified effects of brief photoactivation of Tac1+; CGRP- projections from the PBN to the vIRt (top) and CeA (bottom) on respiratory cycle length relative to the time of stimulation (abscissa and ordinate normalized to mean sham respiratory cycle length). Individual stimulations of the vIRt ($n = 245$ awake, $n = 225$ isoflurane) or CeA ($n = 176$ awake, $n = 198$ isoflurane) during inspiration or expiration are represented by green and blue dots, respectively; sham stimulations are shown in light gray. **h** Latency from photostimulations to the onset of the next breath expressed as a % of sham stimulations in awake or isoflurane anesthetized mice. RM two-way ANOVA with Bonferroni's multiple comparisons tests ($n = 5$ vIRt, $n = 4$ CeA). ****$p < 0.0001$. Means±SE. Source data and statistical details provided in Source Data file.

mechano-feedback, and allow it to adapt to chronic changes in metabolic or environmental demands (e.g.,[23,25,74–78]). Yet, how breathing is integrated with higher-order brain functions, how these neural substrates produce rhythmicity that can transition from being extremely robust and reliable to highly dynamic and labile, and how very rapid breathing patterns are produced have remained significant gaps in our knowledge. Here, we implemented an intersectional genetic approach to identify a subset of PBN neurons characterized by the expression of Tac1 but not CGRP that, when activated in the awake state, drive an abrupt transition to rapid breathing patterns that are distinct from chemo-reflex-driven breathing and exemplify state-dependent respiratory control in mice. Using cell-type- and projection-specific, circuit-dissection strategies to map and manipulate these neurons, we find that their respiratory effects are primarily mediated by direct projections to the vIRt of the medulla, which exert non-canonical, state-dependent, and respiratory phase-independent control of the respiratory rhythm.

The pons in general has a long history as an important region for respiratory control. Early decerebrate and in vitro experiments suggested that the pons contained a "pneumotaxic" center based on surgical transections or more specific lesions of its dorsolateral regions that resulted in elongated inspiratory durations known as apneusis[79–81]. These and many subsequent studies using similar approaches shaped the current view that the primary respiratory functions of the PBN region, including its Kölliker-Fuse subdivision in particular, are regulation of the inspiratory to expiratory phase transition, modulation of expiratory airflow patterns via the control of upper airway patency, and control of respiratory reflexes[27,62]. However, these functional roles have been largely characterized using anesthetized, in situ, or in vitro preparations, which are highly effective for investigating mechanisms of automatic respiratory control, but cannot identify anatomical regions, cell types, or mechanisms that may be conditionally activated in the awake state to regulate breathing.

The PBN is also an important component of the descending pain modulation pathway[82,83], and more recent opto- and chemo-genetic manipulations of PBN neurons in awake mice have revealed that MOR-expressing neurons in this region are important for the integration of breathing with pain and anxiety[17]. However, MOR expression, encoded by *Oprm1*, is widespread in the PBN (~65% of all PBL cells; Fig. 2a) including the KF region[84]; photoactivating these neurons significantly affects breathing in anesthetized mice[17,51], suggesting that they play an important role in automatic respiratory control. This is perhaps most apparent in the context of opioid-induced respiratory depression (OIRD), which emerges through the combined effects of opioids in multiple respiratory regions including the preBötC and PBN[85–90]. Indeed, suppressing the activity of MOR-expressing neurons in the PBN, either with local injections of opioids or via opto- or chemo-genetic inhibition, is sufficient to induce a depression of breathing frequency in both awake or anesthetized mice[17,51,84], indicating that

some of these neurons provide a constitutive excitatory drive to the respiratory CPG that is not gated by state. This is consistent with current respiratory circuit models that include a pontine source of tonic excitatory drive to preBötC neurons[91,92]. Our experiments in awake freely behaving mice identify a small subset of PBL cells (~13%), many of which express *Oprm1*, that exert potent control of breathing that is highly state-dependent. Indeed, when awake, this subset of Tac1-expressing neurons can entrain breathing frequency through the maximum physiological range; but, in contrast to stimulations targeting all *Oprm1* PBN neurons (Fig. S3)[17] or other manipulations of the KF[84,93], activation of Tac1 neurons does not affect breathing in the anesthetized state, suggesting this circuit does not have a significant role in automatic respiratory control.

The PBN is also a key site of convergence for ascending pathways that function as a general alarm system, mediating behavioral and emotional responses to threats[50,94]. These ascending pathways transmit peripheral and visceral aversive sensations to the forebrain via CGRP-expressing neurons in the PBN[64,95]. Here, we find that the respiratory effects of Tac1 PBN neurons are opposed by CGRP-expressing neurons, which promote the state-dependent suppression of rapid breathing patterns. This finding further illustrates the complex functional roles of neuronal populations in the PBN and the importance of cell-type-specific manipulations. Because CGRP PBN neurons relay fear- and pain-related threat signals and promote freezing behavior[64], our results could indicate an opposing role for CGRP and Tac1 PBN neurons in the coordination of breathing during freezing and escape behavioral responses to threats. Alternatively, Tac1+; CGRP- neurons may be conditionally activated to drive sniffing behavior as a general feature of breathing in awake rodents. Indeed, our principle-component analyses demonstrate that breathing waveforms produced by activation of these neurons are very similar to rapid breathing patterns elicited spontaneously in the absence of any overt threat signals. In this context, it is likely that most rapid breathing patterns elicited spontaneously reflect exploratory sniffing, rather than fear, pain, or escape respiratory responses. Further, like sniffs which can abruptly interrupt resting breathing activity, activation of Tac1+; CGRP- neurons could evoke a breath at any time point during the respiratory cycle and often led to a brief "bout" of rapid breathing that outlasted the stimulus. This is also consistent with fiberphotometry recordings of MOR PBN neurons that project to the preBötC region indicating that some of these neurons are activated during sniffing bouts, and that a subset may be specifically involved in the regulation of breathing, but not responses to pain or negative affective states[17,51]. Future studies to inhibit the activity of Tac1+; CGRP- neurons will be important to further define whether this circuit is important for the state-dependent control of breathing in general or if it is specifically recruited under certain behavioral or emotional contexts associated with rapid breathing or other slower breathing patterns.

Consistent with its role in the processing of threat- and pain-related signals, the PBN[96–100] and CGRP neurons in particular[101], are known to contribute to ascending arousal responses to $CO_2$ and/or hypoxia. Indeed, activation of CGRP neurons in the PBN promotes wakefulness, whereas inhibition of these neurons or their ascending projections to the forebrain — primarily the basal forebrain and CeA — delays $CO_2$-induced arousal from sleep[101]. In addition, some evidence suggests that PBN neurons may contribute to the ventilatory response to hypercapnia via activation of forebrain or cortical regions that feedback onto medullary respiratory centers[99]. These ascending feedback loops may contribute to the enhanced respiratory response to $CO_2$ during wakefulness compared to sleep or anesthesia[102]. However, photo-inhibition of parabrachial CGRP neurons does not change ventilatory drive to $CO_2$, suggesting that this ascending CGRP circuit has little, if any, effect on respiratory responses to hypercapnia[101]. Indeed, we find that neither activation of CGRP nor Tac1 PBN neurons elicits changes in breathing that resemble those driven by chemoreflexes. Instead, activation of CGRP PBN neurons promotes a slow regular breathing pattern, whereas Tac1 PBN neuron stimulation produces a breathing pattern that is much faster and distinct from chemoreflex-driven breathing patterns. Further, like Tac1 PBN neurons, activation of CGRP neurons had no effect on respiratory patterns in the anesthetized state, suggesting that CGRP neurons do not significantly contribute to the automatic regulation of breathing. We also found that, despite the known involvement of the CeA in breathing regulation[32,73,103,104] and the arousal response to $CO_2$[101], projections from Tac1+; CGRP- PBN neurons to the CeA play a minor role, if any, in their potent state-dependent control of breathing. Thus, it is unlikely that the CeA neurons that receive inputs from Tac1+; CGRP- PBN neurons directly influence the respiratory CPG.

In contrast to their ascending projections to the CeA, we found that specifically stimulating the descending axonal Tac1+; CGRP- projections within the vIRt of the medulla, which encompasses the preBötC, was sufficient to produce rapid breathing patterns similar to those generated spontaneously. The presence of this direct descending pathway to respiratory rhythm generating regions[32,60] is consistent with the rapid (~40–50 ms) and extremely reliable respiratory responses that were elicited by even brief activation of Tac1+; CGRP- cell bodies or their projections within the vIRt. These stimulations dramatically shortened respiratory-cycle length regardless of whether the stimulus occurred during the inspiratory or expiratory phase, and the timing of evoked breaths was seemingly independent from the ongoing respiratory rhythm. Yet, in the anesthetized state, activation of Tac1+; CGRP- PBN projections in the vIRt no longer had any effects on the breathing rhythm. These findings indicate that the respiratory-phase-independence and state-dependence of this respiratory control circuit is preserved at the level of the vIRt.

These observations are particularly striking given our current understanding of preBötC rhythm-generating properties. Between breaths, the cellular and network properties of glutamatergic rhythm-generating preBötC neurons promote a gradual increase in their excitability until a synchronized burst of activity is produced that drives inspiration[22,105]. Following inspiratory bursts, these neurons become refractory[106] for a period that is proportional to the magnitude of the preceding inspiratory burst[56]. As such, larger bursts result in longer refractory periods, which delays the buildup in excitability that leads to the next burst—thereby slowing breathing frequency[105]. This repeating 3-phase process of building excitability, burst, and refractory period makes the effects of manipulating these rhythmogenic mechanisms highly dependent on the timing of the stimulus relative to the phase of the ongoing rhythm. For example, optogenetic activation of glutamatergic neurons during inspiration causes an augmented inspiratory burst with a long refractory period and delays the onset of the next breath, whereas the same stimulus delivered between breaths can either have little effect or advance the subsequent breath

depending on how much the network has recovered from its refractory period at the time of the stimulus. The effects of manipulating inhibitory GABAergic and/or glycinergic neurons that are integrated within the preBötC network[107,108] are similarly dependent on respiratory phase but in the opposite direction, shortening the respiratory cycle when activated during inspiration and lengthening it when activated between inspirations[56,57]. This respiratory phase-dependence is conserved in awake, anesthetized, or in vitro preparations and is a fundamental property of the mechanisms that automatically generate the breathing rhythm.

Hence, our experiments demonstrating that Tac1+; CGRP- PBN neurons elicit respiratory phase-independent effects at the level of the vIRt suggest that this circuit may drive state-dependent and conditional changes in breathing via activation of non-canonical mechanisms of respiratory-rhythm generation. Indeed, it is becoming increasingly appreciated that the reticular regions of the brainstem contain many interacting circuits capable of producing rhythmicity related to non-respiratory functions such as whisking, chewing, vocalization, and swallowing[109–112], which are primarily active during the awake state. The inspiratory rhythm generated by the preBötC is considered the "master clock" that coordinates these conditional rhythmic behaviors with breathing[113]. However, it is generally assumed that a single oscillator and/or a single mechanism is responsible for producing all forms of the inspiratory rhythm. Based on our results, we propose that there may be distinct mechanisms in the vIRt that underlie generation of the inexorable, automatic inspiratory rhythm and conditional state-dependent inspiratory rhythms such as sniffing.

## Methods

### Subjects

Experiments were conducted on adult (P78-P135) male and female C57Bl/6 J mice. *Tac1Cre* mice were obtained from Jackson Laboratory (Stock #: 021877). *CalcaFLPo* mice were generated at the University of Washington by Richard Palmiter (see below). *CalcaCre* and *Oprm1Cre* mice were generated as described[51,114]. Mice were bred and housed in a temperature- (65-75˚F) and humidity (40-60%)-controlled vivarium on a 12-hr light cycle with access to food and water *ad libitum* at Seattle Children's Research Institute (SCRI) and/or the University of Washington. All procedures were approved by both the University of Washington and the Seattle Children's Research Institute IACUC and were conducted in accordance with the Guide for the Care and Use of Animals in Research.

### Generation of CalcaFLPo:DsRed mouse line

A targeting construct with FLPo-DsRed inserted at the initiation codon for CGRP loxP-flanked SV-Neo for positive selection and HSV-TK and Pgk-Dta for negative selection was prepared as described in Fig. S4. The construct was linearized with Asc1 and electroporated into G4 (C57BL/6 x SJL hybrid) ES cells. ES cells with correct targeting were identified by Southern blot using DNA digested with *BstE*II and a $^{32}$P-labeled probe outside one of the arms; 13 of 42 clones were correctly targeted and most had single inserts based on a second hybridization with a probe for Neo. Positive clones were injected into blastocysts and transferred to pseudo-pregnant females. Mice from one clone that gave germ-line transmission were bred to *MeoxCre* mice to delete the SV-Neo gene. After removing the *MeoxCre* from the genetic background, the *CalcaFLPo:DsRed* mice were continually backcrossed to C57BL/6 mice for >6 generations and then inbred and maintained as homozygotes. A 3-primer strategy was used for routine genotyping;

Calca-F 5' CCA GGT CCA TGG GCT TAT AGA AC;
Calca-R 5' GGA AGT GGT GAA AGC ATT TTG TTA G;
FLPo-R 5' GCA CAG GAT GTC GAA CTG GC

PCR for 34 cycles at 60 °C annealing temperature gives a 350-bp band for wild type and a 150-bp band for targeted allele.

## Viral tools and production

Plasmids for pAAV1-Ef1a-Cre$_{ON}$/FLPo$_{OFF}$-ChR2:eYFP[66] were acquired from AddGene. pAAV-Ef1α-DIO-ChR2:YFP and pAAV-hSyn-DIO-YFP DNA plasmids were provided by K. Deisseroth (Stanford University). pAAV-Ef1α-DIO-Synaptophysin:mCherry DNA plasmid was generated from pAAV-Ef1α-DIO-Synaptophysin:GFP[115]. Viruses were prepared in-house, HEK cells were transfected with each plasmid plus pDG1 (AAV1 coat stereotype) helper plasmid; viruses were purified by sucrose and CsCl-gradient centrifugation steps, re-suspended in 0.1 M phosphate-buffered saline (PBS) at -$10^{13}$ viral particles/ml, aliquoted, and stored at −80 °C.

## Stereotaxic surgeries

Stereotaxic surgeries were performed via standard methods as previously described[64]. Mice were anesthetized with isoflurane (1–4%), treated with ketoprofen (5 mg/kg) for analgesia and mounted in a stereotaxic frame (Kopf Model 1900). Hair overlying the scalp was removed and the exposed skin was prepared via 3 sets of alternating washes with betadine and ethanol. An incision of ~2 cm was made along the midline of the skull from just caudal to the intraocular line to just past the lambdoid sutures. The cranial sutures were visualized via a microscope mounted to the stereotaxic frame and the Bregma to Lambda distance was established along with the leveling of the skull in both the x and y planes. The skull overlying stereotaxic targets, determined with respect to Bregma, was removed via a drill mounted to the stereotaxic frame and fitted with an endmill drill bit. The z coordinate was zeroed on dura overlying the target region and either a glass micropipette filled with virus, or a glass fiber optic was lowered to the appropriate z coordinates to allow the infusion of virus (200 nL over ~1 min then left in place for 5 min before withdrawal from the brain) or the positioning of a fiber optic to deliver light stimulation to the target region. Targeting coordinates were referenced from bregma for AP and ML coordinates and dura for the DV coordinate and were as follows: PBN fibers and virus (AP: −4.8, ML:±1.4, and DV: −3.3 for virus and −3.0 for optical fibers) CeA fibers (AP: −1.22, ML: ± 2.5, DV: −3.75) and pBotC fibers (AP: −6.75, ML:±1.2, and DV: −4.7). An additional hole was drilled in the skull to allow the anchoring of a bone screw (FST Item No. 19101-00). The fiber-optics and the bone screw were cemented to the skull with cyanoacrylate hardened with dental acrylic monomer. A headcap reinforcing all implants and filling the incision was formed from dental acrylic and the ends of the wound were secured with suture (NDC ProAdvantage Item No. P420661). Isoflurane anesthesia was discontinued, and mice were allowed to recover for at least 3 weeks prior to undergoing experiments.

## Whole-body plethysmography

Mice were briefly anesthetized with isoflurane and attached to a bilateral fiber-optic patch cord with a rotary joint integrated within a custom-built barometric plethysmograph chamber. Pressure signals were amplified (Buxco), digitized (Digidata, Axon Instruments 1550 A), and low-pass filtered (0.1 Hz). Data were collected and analyzed using pCLAMP 9.0 software (Molecular Devices). The optical patch cord was connected to a 473-nm DPSSL laser, and power was set such that an estimated 1.5–2.0 mW was produced at the tip of each implanted fiber. Chambers were continuously supplied with gasses at 400 ml/min from premixed tanks containing either room air (21% O$_2$, balance N$_2$), hypercapnia/hypoxia (5% CO$_2$, 10% O$_2$, balance N$_2$) or severe hypercapnia (10% CO$_2$, 21% O$_2$, balance N$_2$) that flowed through an isoflurane vaporizer to allow introduction of isoflurane to the chamber. Prior to starting experiments, alert unanesthetized mice were allowed to acclimate to the plethysmography chamber for at least 60 min. For assessment of automatic and state-dependent breathing patterns (Fig. 1), breathing was first recorded under room air conditions

(30 min), followed by hypoxic hypercapnia (20 min), severe hypercapnia (20 min), and then following induction of light isoflurane anesthesia (~1.5%). The mice were allowed to recover in room air conditions for >20 min between each condition. All exposures occurred sequentially on the same day, between approximately 12 pm and 4 pm. Data was analyzed during the last ~10 min of each condition. Although usually not necessary, mice were prevented from sleeping, as indicated by closed eyes, during the recording period. The onset of rapid breathing bouts was defined as a transition from 5 or more breaths with a frequency <6 Hz followed by 6 or more breaths with a frequency >8 Hz. These six breaths >8 Hz were color coded in magenta to indicate rapid breathing bout onsets in principle component space. Breath peak inspiratory pressure and area were normalized to values under normal resting breathing under room air conditions (slow mode). Frequency and peak inspiratory pressure probability density histograms were binned in 0.25 Hz and 0.05 arbitrary unit increments, respectively. Slow and fast mode breathing frequencies were separated by determining the least probable breathing frequency bin between fast and slow breathing modes for each mouse. Breathing frequencies above and below this value were then averaged to determine mean slow and fast mode frequencies. For 30-Hz and entrainment (3–15 Hz) optogenetic stimulations, each recording session consisted of 5 to 10 45-s trials containing a 10- to 15-s stimulation period. For respiratory phase-resetting experiments, data were recorded during 50–100 30-s trials containing a single 25-ms light pulse. Optogenetic stimulations were performed randomly during wakefulness independent of whether mice were active or resting, and were repeated following ~5 min of isoflurane or ~10 min of urethane anesthesia.

## Surgical procedures for access to the ventral medulla

Mice were induced with isoflurane (~3%), transferred to urethane anesthesia (1.5 g/kg, i.p.), and placed supine on a custom heated surgical table to maintain body temp at -37 °C. The trachea was exposed through a midline incision and cannulated with a curved (180 degree) tracheal tube (24 G) inserted caudal to the larynx. Mice spontaneously breathed 100% O$_2$ throughout the remainder of the surgery and experimental protocol. The trachea and esophagus were removed rostral to the tracheal tube, and the underlying muscles were removed to expose the basal surface of the occipital bone. The portion of the occipital bone and dura overlying the ventral medullary surface were removed, and the exposed surface of the ventral medulla was superfused with warmed (~37 °C) artificial cerebrospinal fluid (aCSF; in mM: 118 NaCl, 3.0 KCl, 25 NaHCO$_3$, 1 NaH$_2$PO$_4$, 1.0 MgCl$_2$, 1.5 CaCl$_2$, 30 D-glucose with an osmolarity of 305–312 mOSM and a pH of 7.40–7.45 when equilibrated with carbogen (95% O$_2$, 5% CO$_2$) at ambient pressure). The hypoglossal nerve (XII) was surgically isolated unilaterally, cut distally, and recorded using a fire polished pulled glass pipette suction electrode containing aCSF. Electrical activity from the XII nerve was amplified (10,000X), filtered (low pass 300 Hz, high pass 5 kHz), rectified, integrated, and digitized (Digidata 1550 A, Axon Instruments). For stimulation of Tac1+ or Tac1+; CGRP- terminals, 200-μm diameter fiber optics, each emitting 2-6 mW of 473-nm light at the fiber tip, were placed bilaterally in light contact with the surface of the medulla at previously defined coordinates that correspond with the location of the preBötC[56].

## Post hoc processing for histology

At the conclusion of experiments, animals were deeply anesthetized with isoflurane and perfused transcardially with ice-cold phosphate-buffered saline (PBS) followed by ice-cold 4% paraformaldehyde in PBS. Brains were removed and stored in 4% PFA at 4 °C overnight and then transferred to 30% sucrose at 4 °C for cryoprotection. Brains were frozen in mounting medium and stored at −80 °C. Using a cryostat, brains were sectioned at 30 μm with each ROI collected in a series of

five sections with one mounted directly to a glass slide and the other four sections stored free-floating in PBS for staining via immunohistochemistry or wet mounting. Sections were examined to confirm that optical fibers were placed in the intended location(s). No mice were excluded from analysis due to off-target fiber positioning. For immunohistochemistry free-floating sections were washed three times in PBS with 0.2% Triton X-100 (PBST) for 5 min before incubation in blocking solution (3% normal donkey serum in PBST) for 1 h at 4 °C. Sections were then incubated overnight in PBS with 3% normal donkey serum and primary antibodies for chicken-anti-GFP (1:10,000, Abcam, ab13970) and, when needed, rabbit-anti-DsRed (1:1000, TaKaRa, Cat No. 632496). Slices were then washed three times in PBS and incubated for 1 h in PBS with secondary antibodies: Alexa Fluor 488 donkey anti-chicken and Alexa Fluor 594 donkey anti-rabbit (1:1000 Jackson ImmunoResearch). Sections were then washed three times in PBS and mounted on glass slides cover slipped and sealed with Fluoromount-G (Southern Biotech). Images were collected on either a Keyence microscope (BZ-X700) or Olympus Fluoview FV-1200 confocal microscope.

### Fluorescent in-situ hybridization

A separate cohort of C57Bl/6J mice were anesthetized with isoflurane and decapitated. Brains were extracted and flash frozen for 1 min in methylbutane cooled to ~−30 °C with dry ice. Brains were stored at −80 °C and then 20-μm sections were collected using a cryostat. Tissue sections were mounted onto glass slides and stored at −80 °C. Sections through the PBN were stained using RNAscope (Fluorescent Multiplex Assay ACD bio) with probes against *Tac1, Calca*, and *Oprm1* mRNA. After staining, slides were imaged on a confocal microscope (Olympus Fluoview FV-1200). Images were analyzed using CellProfiler[116,117] with a custom pipeline (available upon request) based on the ACD bio reference material to determine the number of cells based on DAPI signal and counts of proximal dots of RNAscope signal within the region of interest. Cellular level RNA counts were compiled in Microsoft Excel and then passed into R studio and analyzed via a custom script (available upon request) to determine relationships between RNA expression.

### Data analysis, and visualization

Data analysis was performed using Clampfit 10 (Axon), Excel, GraphPad Prism, MatLab and R Studio. All group data is reported as means ± SE., and significance ($p < 0.05$) was determined using appropriate statistical comparisons and post-hoc multiple comparisons tests (described in Figure Legends and Source Data files). Significance indicated as follows: *$p < 0.05$; **$p < 0.01$; ***$p < 0.001$; ****$p < 0.0001$. ns indicates not significant, $p > 0.05$. Data was visualized and figures were assembled using a combination of Clampfit, GraphPad, ImageJ, and PowerPoint software. A mouse brain atlas[118] was used for anatomical reference.

### Reporting summary

Further information on research design is available in the Nature Portfolio Reporting Summary linked to this article.

## Data availability

All data is generated or analyzed during this study are included within this published article and associated supplementary material or are available from the corresponding author upon request without restriction. Source data are provided with this paper.

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

## Acknowledgements

We thank our funding sources NIH R00-HL145004 (N.A.B.), NIH R01-HL166317 (N.A.B.), NIH R01-DA24908 (R.D.P., A.J.B.), HHMI (R.D.P., J.W.A.) for supporting this work, Jordan L. Pauli for confocal imaging, Susan Phelps for maintaining mouse lines used in this study, and Ryan Phillips for helpful comments on this manuscript. This article is subject to HHMI's Open Access to Publication policy. HHMI lab heads have previously granted a nonexclusive CC BY 4.0 license to the public and a sublicense to HHMI in their research articles. Pursuant to those licenses, the author-accepted manuscript can be made freely available under a CC BY 4.0 license immediately upon publication.

## Author contributions

Conceptualization, N.A.B.; methodology, J.W.A., A.J.B., R.D.P., N.A.B.; investigation, J.W.A., A.J.B., N.A.B.; formal analysis, J.W.A., N.A.B.; writing–original draft, J.W.A., N.A.B.; writing–review and editing, J.W.A., R.D.P., N.A.B.; funding acquisition, N.A.B., R.D.P.; visualization, J.W.A., N.A.B.; supervision, N.A.B., R.D.P.

## Competing interests

The authors declare no competing interests.
