## [Peer Review File · Nature Communications]

Parabrachial tachykinin1-expressing neurons involved in state-dependent breathing controlREVIEWER COMMENTS

Reviewer #1 (Remarks to the Author):

This study was designed to investigate the impact of two neuronal groups (tachykinin1-expressing neurons and calcitonin gene related peptide) in the PBN on the breathing frequency of mice during wakefulness vs. other states/conditions (chemoreflex activation and anesthesia). The authors used a variety of sophisticated and cutting-edge methods in the implementation of a systematic approach that was used to examine the influence of Tac 1 and CGRP neurons on breathing during wakefulness vs. other states. This included (a) using RNA scope to determine the percentage of Tac1 and CGRP neurons that comprise the PBN (b) photostimulating Tac1 neurons and CDGP neurons during different states (wake and anesthesia) and phases of breathing (inspiration vs. expiration) to determine the presence (or absence of modifications and (c) the use of adeno-associated viruses to determine the neuronal projections and connections that mediated the modifications. The findings overall indicated that photostimulation of Tac1 neurons initiated increases in breathing frequency during wakefulness that were similar to frequencies initiated during sniffing behavior during natural breathing. Likewise, resetting of the respiratory phase was also evident during photostimulation during wakefulness. On the other hand, these modifications were not evident during the anesthetic state. Likewise, using cell type and projection specific circuit strategies the authors showed that the effects were mediated primarily by direct projections to the ventral intermediate reticular zone.

(1) The manuscript was well written. However, both the introduction and the discussion could be streamlined by eliminating some of the extraneous information not directly related to the study.

(2) The technical expertise employed in the study was excellent. However, other aspects of the experimental design require additional information. For example, when exploring the rapid dynamic breathing patterns during wakefulness it was unclear whether the data collection was obtained for a similar length of time and at the same time of day. More importantly, it was not clear how arousal conditions were controlled or determined. Specifically, were the animals active or resting quietly. If resting quietly how did the investigators differentiate between quiet rest and sleep. How did the investigators document/determine if some of the changes in breathing frequency were arousal related?

(3) It was unclear how the number of animals selected for a given subdivision of the study was determined. More importantly, it is stated that both male and female mice were used in the study but within the figure legends or text there is no breakdown. There is also not a comparison between sexes, so it is unclear the purpose of including a limited n value for both sexes.

(4) The details regarding the completion of statistics were limited in the text and the significance symbols were not defined in the legends. The figures themselves were dense but for the most part the information provided was necessary. The one aspect of the figures that requires clarity is the pooling of data in the OFF position. If the data was pooled one would anticipate that the data shown for the OFF position (for example 3d) would be exactly the same prior to ON PBNTac1 and ON PBNTac1+CGRP. This

is not the case. Thus, an additional explanation is required.

(5) Some aspects of the methods could be clarified. For example, under “Methods – Whole Body Plethysmography” it is not clear whether the chemoreflex stimuli was administered during wakefulness or anesthesia. In addition, it is stated under surgical procedures that the hypoglossal nerve was isolated and recorded, but why these recordings were made and why the data was not presented requires clarity.

(6) Did repeated bouts of photostimulation lead to any modification in breathing frequency/pattern? In addition, was photostimulation applied during active vs. resting wakefulness. Did photostimulation impact arousal state. Is there a reason why photostimulation was not also administered during chemoreflex activation? In addition, is there a reason why spontaneous breathing patterns and patterns during photostimulation were not determined during natural sleep vs. anesthesia?

(7) Overall, extensive work was completed to explore the role of Tac1 neurons in initiating a specific breathing pattern during wakefulness in mice that was not initiated under a state of anesthesia. On the other hand, the phenomena itself seems to be related more to the mouse and it was less clear if these findings would ultimately have translational significance in humans, particularly when comparing wakefulness to the natural state of sleep.

Reviewer #2 (Remarks to the Author):

The manuscript entitled “Parabrachial tachykinin1-expressing neurons control state-dependent breathing patterns” demonstrates that the Tachykinin1 positive / CGRP negative neuron within the parabrachial nucleus project to the ventrolateral medulla (among other medullary targets) and that upon ectopic excitation in awake animals, they dramatically stimulate breathing, even >10 Hz, mirroring the speed of sniffing. This contrasts with the slowing of breathing when all CGRP neurons are excited. They show that the Tac1+CGRP- medullary axonal projects directly drive breathing and that the breathing rate can be entrained by the frequency of ectopic optogenetic excitation. Surprisingly, this manuscript shows that this robust increase in respiratory rate upon optogenetic excitation does not occur under anesthesia. The experiments are well performed, and the analysis is quantitative and rigorous. In total, this study leads the authors to conclude that the ability of the parabrachial Tac1 neurons to increase respiratory rate is “state” dependent.

While these results are clear, the significance of “state” dependent modulation is overstated, as the two states compared are anesthetized vs awake animals. It is not surprising that a modulator of breathing has differential impacts under anesthesia. Without demonstrating a role for these neurons in increasing the breathing rate in certain behavior contexts (as insinuated for fast breathing during sniffing behavior), the broad interest of these results is not apparent to this reviewer. To make such a connection, studies including loss of function experiments and monitoring Tac1 PBN neural activity in

many behavioral contexts that have different speeds of breathing would need to be conducted, among others.

A main observation is that Tac1 PBN neurons do not modulate breathing under anesthesia, but Oprm1 PBN neurons do (of which Tac1 is a subset). Two simple models that must be excluded are: 1) a technical difference between this study and the Oprm1 study, or 2) the absent respiratory increase under this level of anesthesia is simply do to the optogenetic excitation of fewer brainstem projecting PBN neurons vs Oprm1 (Fig. 2). To increase the confidence that Tac1 PBN neurons cannot modulate breathing under anesthesia at all, the authors should include two control experiments.

1) The inclusion of optogenetic activation of PBN Oprm1 neurons (as a positive control). As discussed on Page 13, Oprm1 PBN neurons have been shown to increase the breathing rate upon optogenetic excitation under anesthesia. This positive control should be included, since it will demonstrate the ability of the author's experimental system to elicit breathing responses under anesthesia. As written, their reported results are "negative" and so a positive control is necessary.

An example for how this might be important is that in Liu et al, under anesthesia, the increase in respiratory rate observed due to optogenetic stimulation was from ~110 bpm to ~115 / 120 bpm. This same increase in rate appears consistent with the initial increase in respiration seen in isoflurane and urethane in Figure 3d (breath traces). These results appear quite similar and if so, would contradict the conclusion of a "state" dependent effect.

2) The inclusion of a dose response curve of light power & frequency of stimulation. As noted above, the reported results are "negative" and so it must be shown that under a complete dose response curve of laser power and frequency that breathing is not modulated under anesthesia. It would not be surprising that the power required would be different in awake v. anesthetized mice.

In the discussion, several statements go well beyond the study and existing literature.

- Page 12, non-canonical mechanisms. A mechanism is never shown in this manuscript.
- Page 14, activation of Tac1 neurons does not impact automatic respiratory control. Without loss of function studies or those noted above, this conclusion is overstated.
- Page 17, Different speeds of breathing rely upon multiple conditional oscillators. Lesions or silencing of the preBötC stops breathing under any behavioral state. Unless the authors are insinuating that other oscillators are recruiting the preBötC, this statement is unclear and oversimplified.

Reviewer #3 (Remarks to the Author):

I found that this very well written paper is addressing a fundamental question on the control of breathing, bringing new and quite provocative results, using original models, which makes this study one

of the most interesting papers on this topic I have had the opportunity to read over the last few years. This study is well thought and relies on creative thinking; it brings evidence that structures outside the medullary preBöttinger complex, believed to be involved in the generation of every respiratory activity, can independently generate breathing cycles akin to those typically observed in certain physiological behavioral states in mice. It is a study which deserves publication as it is an important step towards a better understanding on how breathing is generated in “real” physiological conditions, when awake. Limiting the implications to “sniffing” (of note this fast-breathing pattern may not be always related to sniffing) is a bit sad though, I believe that the authors could be a bit “bolder” in their conclusions.

I have 3 main comments that the authors may want to consider.

1- The authors would agree that they have not shown that Parabrachial tachykinin1-expressing neurons in the pons mediate these very rapid bouts of spontaneous breathing activity observed in awake mice (when sniffing or doing other things associated with very fast breathing. What they showed is that the pattern of activity produced by Parabrachial tachykinin1-expressing neurons has similar characteristics (frequency, effect of sedation, etc..) to the ones produced in some behavioral states in mice when awake. The similarity between these 2 very fast rhythms does not mean that they rely on the same sets of neurons though (sorry to be a bit trivial here). If this is an important question for the authors, then they must show that animals have lost their ability to display a spontaneous very rapid breathing activity after inhibition of these neurons. I would simply clarify this point in the abstract and the discussion (whether the authors agree or not with my comment). The authors should make clear however that whether there is the link between the 2 rhythms, even if not proven, this does not change their conclusion, preventing any red herring in potential future debates.

2- The ability of Parabrachial tachykinin1-expressing neurons, in contrast to preBöttinger neurons, to generate a very fast breathing rhythm is at the core of the rationale of this study. The authors quote several papers supporting the inability of preBöttinger neurons to produce a respiratory rhythm that would reach ~ 13 Hz (peak of the spontaneous frequency in mice). The question the authors should in my opinion briefly discuss is therefore the following: can Parabrachial tachykinin1-expressing neurons also generate breathing activity at a slower rate (there is, from what I understand, entrainment of the respiratory rhythm at 6 Hz, what about lower frequencies?). If true, the authors should at least discuss the possibility that Parabrachial tachykinin1-expressing neurons could (might) be involved in slower breathing patterns beyond those associated with sniffing.

3- I realize that the authors do not study breathing control in humans, yet they should at least mention some of more than 50 year literature that has tried to determine how volitional breathing (volitionally generated breaths) operate in humans. I have included below a list of references that the authors may find relevant to this question. Possible clinical implications, which are a bit different from the question of the very fast rhythm generated in mice, could be briefly mentioned in the discussion? What do you think?

PH

D.R. Corfield, K. Murphy, A. Guz Does the motor cortical control of the diaphragm ‘bypass’ the brain stem respiratory centres in man *Respir. Physiol.*, 114 (1998), pp. 109-117

P. Haouzi, B. Chenuel, B.J. Whipp

Control of breathing during cortical substitution of the spontaneous automatic respiratory rhythm
Respiratory Physiology & Neurobiology 159, 2007, Pages 211-218

S.C. Gandevia, J.C. Rothwell Activation of the human diaphragm from the motor cortex *J. Physiol.*, 384 (1987), pp. 109-118

F. Plum Neurological integration of behavioural and metabolic control of breathing R. Porter, J.A. Churchill (Eds.), *Breathing: Hering-Breuer Centenary Symposium*, Ciba Foundation, London (1970), pp. 159-181

M.J. Aminoff, T.A. Sears Spinal integration of segmental, cortical and breathing inputs to thoracic respiratory motoneurons
J. Physiol., 215 (1971), pp. 557-575

Reviewer #1 (Remarks to the Author):

This study was designed to investigate the impact of two neuronal groups (tachykinin1-expressing neurons and calcitonin gene related peptide) in the PBN on the breathing frequency of mice during wakefulness vs. other states/conditions (chemoreflex activation and anesthesia). The authors used a variety of sophisticated and cutting-edge methods in the implementation of a systematic approach that was used to examine the influence of Tac 1 and CGRP neurons on breathing during wakefulness vs. other states. This included (a) using RNA scope to determine the percentage of Tac1 and CGRP neurons that comprise the PBN (b) photostimulating Tac1 neurons and CDGP neurons during different states (wake and anesthesia) and phases of breathing (inspiration vs. expiration) to determine the presence (or absence of modifications and (c) the use of adeno-associated viruses to determine the neuronal projections and connections that mediated the modifications. The findings overall indicated that photostimulation of Tac1 neurons initiated increases in breathing frequency during wakefulness that were similar to frequencies initiated during sniffing behavior during natural breathing. Likewise, resetting of the respiratory phase was also evident during photostimulation during wakefulness. On the other hand, these modifications were not evident during the anesthetic state. Likewise, using cell type and projection specific circuit strategies the authors showed that the effects were mediated primarily by direct projections to the ventral intermediate reticular zone.

(1) The manuscript was well written. However, both the introduction and the discussion could be streamlined by eliminating some of the extraneous information not directly related to the study.

Thank you for the positive comments and thoughtful critique of our manuscript. We have revised the INTRODUCTION and DISCUSSION with this comment in mind.

(2) The technical expertise employed in the study was excellent. However, other aspects of the experimental design require additional information. For example, when exploring the rapid dynamic breathing patterns during wakefulness it was unclear whether the data collection was obtained for a similar length of time and at the same time of day. More importantly, it was not clear how arousal conditions were controlled or determined. Specifically, were the animals active or resting quietly. If resting quietly how did the investigators differentiate between quiet rest and sleep. How did the investigators document/determine if some of the changes in breathing frequency were arousal related?

Thank you for pointing this out. We have added clarifications to METHODS to address these important comments. Following a 60 min period of acclimation to the plethysmography chamber, breathing was measured for 30 min in control room air conditions, followed by 20 min in hypoxic hypercapnia, then severe hypercapnia, and then isoflurane (room air). The mice were allowed to recover in room air conditions for >20 min between each condition. For each mouse, all exposures occurred sequentially on the same day, between approximately noon and 4pm. Data was analyzed during the last 10 min of each condition. Although usually not necessary, mice

were prevented from sleeping (indicated by closed eyes) during the recording period. Thus, the breathing patterns characterized in this experiment represent the “awake state”. However, beyond this, mice were allowed to behave freely, and all breaths were included in the analysis independent of whether the mice were active or resting. Indeed, our intent was to look at awake breathing patterns as a whole, including modified breathing patterns associated with active behaviors. Therefore, we did not separate active vs. resting stimulations. We fully agree with the reviewer that the level of arousal during the awake state can be nuanced and is likely to have effects on breathing pattern. In fact, we suspect that the Tac1 PBN neurons described here might represent one pathway that links changes in arousal state to changes in breathing, and we are working on experiments to record the activity of these neurons across different behavioral and arousal states to begin to understand this in greater detail.

(3) It was unclear how the number of animals selected for a given subdivision of the study was determined. More importantly, it is stated that both male and female mice were used in the study but within the figure legends or text there is no breakdown. There is also not a comparison between sexes, so it is unclear the purpose of including a limited n value for both sexes.

We apologize for the confusion. Based on the magnitude and consistency of the effects observed in pilot experiments, as well as previous studies involving similar experiments, we expected n=5 per group would provide sufficient statistical power. However, group sizes were sometimes larger if the litters produced for a given experiment contained more than 5 pups.

The purpose of including both sexes is primarily to utilize all available mice. These transgenic animals are bred in house and require a lot of time and resources to produce and maintain. We did not do a quantitative analysis of the data based on sex, because as the reviewer points out, there would be a limited n for each group. However, the effects we observed during stimulation of Tac1 or CGRP PBN neurons were dramatic and very consistent across animals, with no qualitative differences between males and females. Thus, in this case, our data did not reveal any sex-dependent trends that would warrant doubling our group sizes to allow for this statistical analysis.

(4) The details regarding the completion of statistics were limited in the text and the significance symbols were not defined in the legends. The figures themselves were dense but for the most part the information provided was necessary. The one aspect of the figures that requires clarity is the pooling of data in the OFF position. If the data was pooled one would anticipate that the data shown for the OFF position (for example 3d) would be exactly the same prior to ON PBNTac1 and ON PBNTac1+CGRP. This is not the case. Thus, an additional explanation is required.

Thanks for pointing this out. We now define significance symbols in both **FIGURE LEGENDS** and in the “Data Analysis” section of **METHODS**. All specific details regarding statistical tests are now included in **FIGURE LEGENDS** and/or the included source data files.

The pooling of data for the OFF condition only applies to the probability density histograms that illustrate the distribution of breath frequencies/pressures during each condition, as indicated in

FIGURE LEGENDS. Data from the control condition for both PBN^{Tac1} and PBN^{CGRP} mice (i.e. Light OFF) was pooled in this case to improve the clarity of the figure. However, data was not pooled in the associated graphs below showing the mean value for each replicate and the overall mean \pm SE with statistics to compare paired light ON and light OFF conditions.

(5) Some aspects of the methods could be clarified. For example, under “Methods – Whole Body Plethysmography” it is not clear whether the chemoreflex stimuli was administered during wakefulness or anesthesia. In addition, it is stated under surgical procedures that the hypoglossal nerve was isolated and recorded, but why these recordings were made and why the data was not presented requires clarity.

We have revised these sections to clarify that the chemoreflex stimuli were administered in the awake freely behaving state, prior to induction of isoflurane anesthesia. The data involving hypoglossal nerve recording are shown as a supplemental figure (Fig 6 supplement 3 and 4). The purpose of these experiments was to further test the state-dependence of this effect as described in RESULTS line ~323: “In some mice, the ventral surface of the medulla was surgically exposed under urethane anesthesia to allow stronger photostimulations of Tac1+; CGRP- neuronal terminals in the vIRt (200- μ m diameter fibers, ~6 mW each), which also failed to elicit changes in breathing (Fig. 6 supplement 3).

(6) Did repeated bouts of photostimulation lead to any modification in breathing frequency/pattern? In addition, was photostimulation applied during active vs. resting wakefulness. Did photostimulation impact arousal state. Is there a reason why photostimulation was not also administered during chemoreflex activation? In addition, is there a reason why spontaneous breathing patterns and patterns during photostimulation were not determined during natural sleep vs. anesthesia?

Thanks for the interesting question regarding potential effects from repeated bouts of stimulation. We have added additional analysis (Fig 3 supplement 4) comparing breathing between the first, second, and third bout of 30-Hz (15 sec long) stimulations. We found no differences between trials during the light OFF or light ON conditions, suggesting neither the magnitude of the stimulation effect nor the intervening breathing patterns changed significantly over the course of multiple trials.

Photostimulations were performed during wakefulness at random (i.e. independent of whether the mouse was active or resting.) As mentioned above, an important aspect of our study was examination of awake breathing patterns as a whole; therefore, we did not specifically apply stimulations to resting vs active conditions. We have added this information to METHODS ~Line 586.

These studies were designed without expectation of the observed dramatic state-dependent effects. Accordingly, stimulations were not specifically delivered during sleep. However, preliminary observations that will form the basis for a follow up study examining in more detail how these neurons may contribute to the link breathing and arousal suggest that stimulation of Tac1 PBN neurons promotes arousal from sleep. This is consistent with activation of CGRP

PBN neurons, which are known to promote arousal and wakefulness, particularly in response to CO₂. This important point is mentioned in DISCUSSION ~ Line 415. Accordingly, we expect it would be quite difficult to assess how/whether activation of these PBN populations affect breathing during natural sleep.

(7) Overall, extensive work was completed to explore the role of Tac1 neurons in initiating a specific breathing pattern during wakefulness in mice that was not initiated under a state of anesthesia. On the other hand, the phenomena itself seems to be related more to the mouse and it was less clear if these findings would ultimately have translational significance in humans, particularly when comparing wakefulness to the natural state of sleep.

Thank you for the thoughtful review. As you suggest, this is a basic science study and although it provides important insights about the control breathing in mice, it remains unknown what the translational or clinical significance of these findings will be. This is true for a lot of what we understand about the neuronal control of breathing, since most research has relied on invasive approaches in rodent or cat model systems.

For our study, mice were an ideal model system to explore state dependent respiratory control because they exhibit breathing patterns that are uniquely produced in the awake state and can be easily identified. In other animals that don't regularly explore their environment through sniffing, including humans, the identification of breathing patterns linked to state-dependent respiratory control mechanisms may be less obvious. Nevertheless, these animals exhibit state-dependent breathing control, and our findings in mice establish a foundation for exploring this further in other model systems and in contexts other than "sniffing". Indeed, as mentioned in the text (~Line 408) it is unknown whether breathing patterns driven by Tac1 PBN neurons exclusively represent sniffing, or if they may be important for regulating state-dependent breathing patterns in general. We feel this represents an exciting avenue for continued research that may one day reveal how the state-dependent interactions between breathing, behavior, and emotion are regulated by the brain, and how they may become dysregulated in the context of e.g. panic disorders or hyperventilation syndromes.

Reviewer #2 (Remarks to the Author):

The manuscript entitled "Parabrachial tachykinin1-expressing neurons control state-dependent breathing patterns" demonstrates that the Tachykinin1 positive / CGRP negative neuron within the parabrachial nucleus project to the ventrolateral medulla (among other medullary targets) and that upon ectopic excitation in awake animals, they dramatically stimulate breathing, even >10 Hz, mirroring the speed of sniffing. This contrasts with the slowing of breathing when all CGRP neurons are excited. They show that the Tac1+CGRP- medullary axonal projects directly drive breathing and that the breathing rate can be entrained by the frequency of ectopic optogenetic excitation. Surprisingly, this manuscript shows that this robust increase in respiratory rate upon optogenetic excitation does not occur under anesthesia. The experiments are well performed, and the analysis is quantitative and rigorous. In total, this study leads the

authors to conclude that the ability of the parabrachial Tac1 neurons to increase respiratory rate is “state” dependent.

While these results are clear, the significance of “state” dependent modulation is overstated, as the two states compared are anesthetized vs awake animals. It is not surprising that a modulator of breathing has differential impacts under anesthesia. Without demonstrating a role for these neurons in increasing the breathing rate in certain behavior contexts (as insinuated for fast breathing during sniffing behavior), the broad interest of these results is not apparent to this reviewer. To make such a connection, studies including loss of function experiments and monitoring Tac1 PBN neural activity in many behavioral contexts that have different speeds of breathing would need to be conducted, among others.

We understand the reviewer’s hesitation regarding the term “state-dependent”, as this terminology gets used broadly. Indeed, “state” is ambiguous and can be used in reference to a change in conditions as dramatic as the transition from in vivo to ex vivo, or as nuanced as a change in the contextual cues in a mouse’s home cage. Moreover, “dependence” in state-dependence can often refer to quite subtle changes in the response to a stimulus when delivered in different “states”. This is not the case for the “state-dependence” we describe here.

Indeed, we don’t share the reviewers view that the effects of anesthesia on the ability of Tac1 PBN and CGRP PBN neurons to alter breathing are “not surprising”. Not only are the effects on breathing in the awake state extremely potent compared to what has been demonstrated previously, but they are almost completely eliminated in modest anesthesia, even during strong 30-Hz stimulations. Our respiratory phase resetting experiments also clearly demonstrate this striking state-dependence. We found this quite surprising, and it is unexpected given what we and others have found in response to manipulation of other respiratory neurons, which generally exert similar or even stronger effects under anesthesia vs. when awake. (e.g. Line ~174).

We fully agree with the reviewer that “loss of function experiments and monitoring Tac1 PBN neural activity in many behavioral contexts that have different speeds of breathing” are important experiments. Indeed, during preparation of this manuscript, we were excited to see fiberphotometry data published demonstrating that a subset *Oprm1* neurons in the PBN, which includes the smaller Tac1 population, has activity that is correlated with spontaneous bouts of rapid breathing (Liu et al., 2021), which nicely complements our findings and conclusions. More specific neuronal recording (Neuropixel) and loss of function experiments targeting Tac1 or CGRP PBN neurons under the numerous behavioral contexts associated with rapid breathing in mice are components of Aims 2 and 3 of our grant that will allow us to continue to pursue this project. However, we feel these experiments are beyond the scope of the current study which already provides a number of important insights about the control of breathing.

A main observation is that Tac1 PBN neurons do not modulate breathing under anesthesia, but *Oprm1* PBN neurons do (of which Tac1 is a subset). Two simple models that must be excluded are: 1) a technical difference between this study and the *Oprm1* study, or 2) the absent respiratory increase under this level of anesthesia is simply do to the optogenetic excitation of fewer brainstem projecting PBN neurons vs *Oprm1* (Fig. 2). To increase the confidence that

Tac1 PBN neurons cannot modulate breathing under anesthesia at all, the authors should include two control experiments.

There are multiple studies using various methodologies that suggest PBN Oprm1 neurons can alter breathing in the anesthetized state. For example, beyond the recent optogenetic activations (Liu et al., 2021), suppression of these neurons with local administration of opioids depresses breathing frequency (OIRD) in the anesthetized state (e.g. Levitt et al., 2015). Thus, some of these neurons continue to provide an excitatory respiratory drive under anesthesia, and accordingly increasing their activity would be expected to increase breathing frequency in the anesthetized state. Similarly, stimulation of PBN neurons in general (of which a large fraction are *Oprm1+*) has been shown to facilitate or sometimes suppress breathing in anesthetized rats (e.g. Chamberlin and Saper 1994).

Although we understand the reviewers point that “the absent respiratory increase under this level of anesthesia is simply due to the optogenetic excitation of fewer brainstem projecting PBN neurons vs Oprm1”, it is notable that (to our surprise) this logic did not apply to the Tac1 population. In this case, activation of all Tac1 PBN neurons had a weaker effect on breathing than activation of the smaller population of Tac1+; CGRP- PBN neurons, not because our stimulations were “weaker” (laser power or number of brainstem projecting neurons) but because Tac1 and CGRP populations have opposing functional roles and Tac1+; CGRP- stimulation is simply more specific for driving rapid breathing patterns. We feel the same logic likely applies to the broader Oprm1 population which contains subpopulations with opposing functions (e.g. Tac1 and CGRP).

1) The inclusion of optogenetic activation of PBN Oprm1 neurons (as a positive control). As discussed on Page 13, Oprm1 PBN neurons have been shown to increase the breathing rate upon optogenetic excitation under anesthesia. This positive control should be included, since it will demonstrate the ability of the author’s experimental system to elicit breathing responses under anesthesia. As written, their reported results are “negative” and so a positive control is necessary.

We now include a cohort of Oprm1 mice, which is included as (Fig. 2 Supplement 2). In 5 of 8 mice, breathing was facilitated, whereas in the other 3 it was suppressed. These differential effects may reflect the broader topographic localization of MOR neurons across PBN subregions (Pauli et al., 2022; Chamberlin and Saper 1994) and/or the opposing functional roles of Tac1+ and CGRP+ subpopulations. Importantly, in all 8 mice, obvious effects on breathing persisted following induction of anesthesia, but were more modest than in the awake state. We expect this is due to loss of the state-dependent contributions of Tac1 and CGRP neurons. These results support the idea that the PBN contains multiple subpopulations of neurons that differentially contribute to both the automatic and state-dependent control of breathing.

An example for how this might be important is that in Liu et al, under anesthesia, the increase in respiratory rate observed due to optogenetic stimulation was from ~110 bpm to ~115 / 120 bpm. This same increase in rate appears consistent with the initial increase in respiration seen in

isoflurane and urethane in Figure 3d (breath traces). These results appear quite similar and if so, would contradict the conclusion of a “state” dependent effect.

In our view, this does not contradict the conclusion of a state-dependent effect, but simply suggests that the state-dependence of Oprm1 PBN neuron activation was not emphasized in the paper by Liu et al. As mentioned above, our results provide a straightforward explanation for why more modest effects of Oprm1 stimulation would be expected in the anesthetized state (i.e. loss of the state-dependent contributions of Tac1 and CGRP neurons). Oprm1 is broadly expressed in the PBN and, as alluded to above, the available data from this study and others suggests that these Oprm1 neurons are not a functionally uniform population. Instead, some Oprm1 neurons provide a constitutive excitatory respiratory drive, while others (Tac1+; CGRP-) provide state-dependent (i.e. conditional) respiratory drive. If this is the case, one would predict that activation of all Oprm1 neurons would increase breathing frequency when awake or anesthetized but with larger effects when activated in the awake state. Although not a focus of their study, this seems to agree with what we find (Fig. 2 supplement 2) and what was observed by Liu et al.

2) The inclusion of a dose response curve of light power & frequency of stimulation. As noted above, the reported results are “negative” and so it must be shown that under a complete dose response curve of laser power and frequency that breathing is not modulated under anesthesia. It would not be surprising that the power required would be different in awake v. anesthetized mice.

We now include analysis of 10-, 20- 30-Hz stimulations in the awake state (Fig. 3 supplement 3). As the reviewer suggests, these “dose response” experiments were initially performed to determine the strength of stimulation that saturated the effect on breathing. As shown in this analysis, 10-, 20-, and 30-Hz stimulations produced similar responses, indicating that 30-Hz stimulation is well beyond the “dose” required to achieve maximal effects in the awake state. Indeed, 30 Hz was chosen to compare effects in the anesthetized vs. awake state specifically because it was a hypersaturating “dose” and would most convincingly demonstrate the state-dependence of the respiratory response, since *even* at 30-Hz, effects on breathing were absent or very nearly absent in the anesthetized state.

Further, to address this in the original submission, we performed the experiments described in Fig. 6 supplement 3, which show that even when using ~3x the laser power (6 mW), 30-Hz stimulation of Tac1+; CGRP- projections in the vIRt has no effect on breathing. To emphasize this, we now include additional dose response analysis comparing 2 mW (as used in the awake state), 4 mW, and 6 mW *continuous* stimulations of Tac1 projections in the vIRt, which also revealed no changes in breathing at any laser power (Fig. 6 supplement 3 and 4). If anything, breathing frequency tends to decline with higher laser powers, consistent with the non-specific effects we have previously observed in control mice that do not express ChR2 (Baertsch et al., 2018, Supplementary Fig. 9).

The reviewer is correct that *in the anesthetized state* our results are “negative”. However, in our view, our results in the awake state serve as the ideal “positive” control, since in this case the same stimulus delivered to the same animal revealed extremely potent effects on breathing.

In the discussion, several statements go well beyond the study and existing literature.

- Page 12, non-canonical mechanisms. A mechanism is never shown in this manuscript.

We have rephrased this statement; however, we are unsure how the reviewer defines “mechanism”. In our view, a “mechanism” is simply how a phenomenon works, which can be dissected at different levels. For example, one could describe a circuit mechanism, network mechanism, cellular, molecular, or biophysical mechanism. To clarify, what we are trying to emphasize here is how the respiratory phase-independence of the Tac1 PBN stimulations differs from the highly phase-dependent effects of manipulations of preBötC neurons. The cellular and network “mechanisms” that underlie the phase-dependent effects of these preBötC manipulations are relatively well understood and are related to properties associated with rhythmogenesis (refractory period, percolation, I_{NaP} activation and inactivation dynamics). Thus, our finding that a similar phase-dependence does not apply to the effects of Tac1 PBN stimulations implies that these neurons drive breathing in a way that is different from what is currently understood about how the respiratory rhythm is generated - i.e. via “non-canonical mechanisms”.

- Page 14, activation of Tac1 neurons does not impact automatic respiratory control. Without loss of function studies or those noted above, this conclusion is overstated.

The full statement in our original submission was “activation of Tac1 neurons does not affect breathing in the anesthetized state, *suggesting* this circuit does not have a significant role in automatic respiratory control.” We agree with the reviewer that loss of function studies will help confirm this suggestion, and we are currently working on these experiments. However, we feel proposing this suggestion based on our activation experiments is not overstated since it is hard to imagine a scenario where activating neurons would have no effect, but suppressing the same neurons would significantly alter breathing.

- Page 17, Different speeds of breathing rely upon multiple conditional oscillators. Lesions or silencing of the preBötC stops breathing under any behavioral state. Unless the authors are insinuating that other oscillators are recruiting the preBötC, this statement is unclear and oversimplified.

We fully agree that acutely silencing the preBötC would be expected to stop breathing under any behavioral state. We do not propose that Tac1 PBN neurons are rhythmogenic, however we also do not exclude this possibility. Indeed, it is feasible that under certain conditions the preBötC could act as a relay for rhythmic activity generated by Tac1 PBN neurons (or elsewhere). In this scenario, the preBötC would still be “necessary” for breathing. However, what we believe is most probable, and what we are speculating here, is that the preBötC and vIRt as a region may have multiple rhythm generating mechanisms that differentially contribute

to automatic vs state-dependent breathing patterns. In this scenario, the preBötC is still necessary for all forms of breathing, but its properties of rhythm and pattern generation are more complicated than our current understanding, which is largely based on insights from in vitro or anesthetized preparations.

Reviewer #3 (Remarks to the Author):

I found that this very well written paper is addressing a fundamental question on the control of breathing, bringing new and quite provocative results, using original models, which makes this study one of the most interesting papers on this topic I have had the opportunity to read over the last few years. This study is well thought and relies on creative thinking; it brings evidence that structures outside the medullary preBötzing complex, believed to be involved in the generation of every respiratory activity, can independently generate breathing cycles akin to those typically observed in certain physiological behavioral states in mice. It is a study which deserves publication as it is an important step towards a better understanding on how breathing is generated in “real” physiological conditions, when awake. Limiting the implications to “sniffing” (of note this fast-breathing pattern may not be always related to sniffing) is a bit sad though, I believe that the authors could be a bit “bolder” in their conclusions.

Thank you for the very kind comments. We agree that these surprising findings are an important step towards a better understanding of the control of breathing in the awake, normally behaving state. We also agree that limiting our conclusions to “sniffing” would be an oversimplification. Indeed, we tried to use the more general terminology “rapid breathing patterns” throughout most of the manuscript for this very reason. However, we also felt it was worthwhile to include some discussion of “sniffing” because it is a clear example of a behavioral modification of breathing and likely reflects most of the rapid spontaneous breathing we used as a comparison here. As mentioned above in response to the other reviewers, we are beginning experiments that will help more specifically define the behavioral and/or emotional states that this PBN circuit is involved in. We greatly appreciate the enthusiasm to be bolder in our conclusions! However, we feel we have achieved reasonable middle ground (see comments from reviewer 2) that does not over- or under-interpret our results. Our ongoing work in this area may justify some of the “bolder” conclusions you suggest in subsequent publications.

I have 3 main comments that the authors may want to consider.

1- The authors would agree that they have not shown that Parabrachial tachykinin1-expressing neurons in the pons mediate these very rapid bouts of spontaneous breathing activity observed in awake mice (when sniffing or doing other things associated with very fast breathing. What they showed is that the pattern of activity produced by Parabrachial tachykinin1-expressing neurons has similar characteristics (frequency, effect of sedation, etc..) to the ones produced in some behavioral states in mice when awake. The similarity between these 2 very fast rhythms does not mean that they rely on the same sets of neurons though (sorry to be a bit trivial here). If this is an important question for the authors, then they must show that animals have lost their

ability to display a spontaneous very rapid breathing activity after inhibition of these neurons. I would simply clarify this point in the abstract and the discussion (whether the authors agree or not with my comment). The authors should make clear however that whether there is the link between the 2 rhythms, even if not proven, this does not change their conclusion, preventing any red herring in potential future debates.

We fully agree with this comment and had this in mind while writing the original manuscript and this revised version. Our main conclusion is that: ~Line 331: "...these results indicate that Tac1+; CGRP- neurons in the PBN can exert potent state-dependent and respiratory phase-independent control of breathing primarily via direct descending projections to the medullary vIRt." ~Line 345 "...we find that their respiratory effects are primarily mediated by direct projections to the vIRt of the medulla, which exert non-canonical state-dependent and respiratory phase-independent control of the respiratory rhythm". These conclusions remain valid even in the unlikely scenario that silencing experiments reveal no deficits in spontaneous rapid breathing patterns. As you suggest, we do not make definitive conclusions about what specific contexts this circuit becomes activated because we haven't yet performed the silencing experiments, which will form that basis of a follow up publication(s). Indeed, we state in the discussion: ~Line 408 "Future studies to inhibit the activity of Tac1+; CGRP- neurons will be important to further define whether this circuit is important for the state-dependent control of breathing in general or if it is specifically recruited under certain behavioral or emotional contexts associated with rapid breathing."

2- The ability of Parabrachial tachykinin1-expressing neurons, in contrast to preBötzinger neurons, to generate a very fast breathing rhythm is at the core of the rationale of this study. The authors quote several papers supporting the inability of preBötzinger neurons to produce a respiratory rhythm that would reach ~ 13 Hz (peak of the spontaneous frequency in mice). The question the authors should in my opinion briefly discuss is therefore the following: can Parabrachial tachykinin1-expressing neurons also generate breathing activity at a slower rate (there is, from what I understand, entrainment of the respiratory rhythm at 6 Hz, what about lower frequencies?). If true, the authors should at least discuss the possibility that Parabrachial tachykinin1-expressing neurons could (might) be involved in slower breathing patterns beyond those associated with sniffing.

Thanks for this interesting comment. We now mention this possibility in the DISCUSSION. We did perform entrainment stimulations starting at 3 Hz and increasing to 15 Hz in all mice. This data is summarized in Fig 4 e-g, and we now include an additional Fig supplement (Fig 4 supplement 1) showing representative traces during these slower stimulation frequencies (3-6 Hz). As stated in the text ~Line 250 "At slower stimulation frequencies (<7 Hz), breathing rate was often increased such that it outpaced the stimulus (Fig. 4b, e; Fig. 4 supplement 1a). Indeed, even single 20-ms light pulses often induced an increase in breathing frequency that outlasted the initial respiratory phase advance (see Fig. 3g; Fig. 4 supplement 1b), indicative of hysteresis in the underlying mechanisms that mediate the changes in breathing elicited by Tac1+; CGRP- neurons." Because even a single short stimulation could evoke a "bout" of rapid breaths, we suspect that the activity of this Tac1 circuit is not likely involved in slower breathing

patterns, at least those < ~7 Hz, since this was about the slowest breathing frequency achieved during Tac1+; CGRP- PBN neurons stimulation.

3- I realize that the authors do not study breathing control in humans, yet they should at least mention some of more than 50 year literature that has tried to determine how volitional breathing (volitionally generated breaths) operate in humans. I have included below a list of references that the authors may find relevant to this question. Possible clinical implications, which are a bit different from the question of the very fast rhythm generated in mice, could be briefly mentioned in the discussion? What do you think?

We now include a few of the suggested citations and agree with the reviewer that our findings in mice have interesting parallels with the literature on volitional breathing in humans. However, considering Reviewer #1's suggestion to streamline the introduction and discussion, we are hesitant to add additional speculation about translational or clinical implications at this time. We hope to write a review on automatic vs. state-dependent respiratory control in the near future, and a summary of the excellent work that has been done on the volitional control of breathing in humans vs. circuit mechanisms in mice will surely be a focus.

PH

D.R. Corfield, K. Murphy, A. Guz Does the motor cortical control of the diaphragm 'bypass' the brain stem respiratory centres in man *Respir. Physiol.*, 114 (1998), pp. 109-117

P. Haouzi, B. Chenuel, B.J. Whipp

Control of breathing during cortical substitution of the spontaneous automatic respiratory rhythm *Respiratory Physiology & Neurobiology* 159, 2007, Pages 211-218

S.C. Gandevia, J.C. Rothwell Activation of the human diaphragm from the motor cortex *J. Physiol.*, 384 (1987), pp. 109-118

F. Plum Neurological integration of behavioural and metabolic control of breathing R. Porter, J.A. Churchill (Eds.), *Breathing: Hering-Breuer Centenary Symposium*, Ciba Foundation, London (1970), pp. 159-181

M.J. Aminoff, T.A. Sears Spinal integration of segmental, cortical and breathing inputs to thoracic respiratory motoneurons

J. Physiol., 215 (1971), pp. 557-575

REVIEWERS' COMMENTS

Reviewer #1 (Remarks to the Author):

I have no additional comments.

Reviewer #2 (Remarks to the Author):

No additional requests.

Reviewer #3 (Remarks to the Author):

The authors have adequately responded to my previous comments. I have not more comments or queries.